# Staggered structural dynamic-mediated selective adsorption of H₂O/D₂O on flexible graphene oxide nanosheets

Ryusuke Futamura [1,2], Taku Iiyama[1,2], Takahiro Ueda [3], Patrick A. Bonnaud[4], François-Xavier Coudert [5], Ayumi Furuse [2], Hideki Tanaka [2], Roland J. -M. Pellenq[6] & Katsumi Kaneko [2] ✉

Graphene oxide (GO) is the one of the most promising family of materials as atomically thin membranes for water-related molecular separation technologies due to its amphipathic nature and layered structure. Here, we show important aspects of GO on water adsorption from molecular dynamics (MD) simulations, in-situ X-ray diffraction (XRD) measurements, and ex-situ nuclear magnetic resonance (NMR) measurements. Although the MD simulations for GO and the reduced GO models revealed that the flexibility of the interlayer spacing could be attributed to the oxygen-functional groups of GO, the ultra-large GO model cannot well explain the observed swelling of GO from XRD experiments. Our MD simulations propose a realistic GO interlayer structure constructed by staggered stacking of flexible GO sheets, which can explain very well the swelling nature upon water adsorption. The transmission electron microscopic (TEM) observation also supports the non-regular staggered stacking structure of GO. Furthermore, we demonstrate the existence of the two distinct types of adsorbed water molecules in the staggered stacking: water bonded with hydrophilic functional groups and "free" mobile water. Finally, we show that the staggered stacking of GO plays a crucial role in H/D isotopic recognition in water adsorption, as well as the high mobility of water molecules.

Water is ubiquitous on our planet but it is urgently important to develop a better control over the water cycle for our sustainability within this century. Graphene oxide (GO) is the one of the most promising family of materials based on graphene, as atomically thin membranes for water-related molecular separation due to their unique properties[1–7]. In particular, the stacked GO layers can uniquely offer nanoscale interlayer spaces for many applications. Nair et al. reported ultrahigh permeance of water molecules on GO membranes which do not permeate any other molecules, including helium[8]. The anomalous behavior of water in GO is linked to the observed high performance of GO membranes as filters not only for water desalination[9] but also for H₂O and D₂O separation[10–12].

A lot of molecular dynamics (MD) simulation studies provided us the microscopic information for anomalous behaviors of water in GO interlayer spaces from both a kinetic and static point of view[11,13–17]. Mouhat et al. showed that the semi-ordered GO sheet with correlated

[1]Department of Chemistry, Faculty of Science, Shinshu University, 3-1-1, Asahi, Matsumoto 390-8621, Japan. [2]Research Initiative for Supra-Materials, Shinshu University, 4-17-1, Wakasato, Nagano 380-8553, Japan. [3]Faculty of Science, Osaka University, 1-13, Machikaneyamacho, Toyonaka 560-0043, Japan. [4]Institute for Materials Research, Tohoku University, Katahira 2-1-1, Aoba, Sendai 980-8577, Japan. [5]Chimie Paris Tech, PSL University, CNRS, Institut de Recherche de Chimie Paris, 11 Pierre and Marie Curie, 75231 Paris, France. [6]European Institute of Membranes (IEM), CNRS and the University of Montpellier, 300 Avenue du Professeur Jeanbrau, 34090 Montpellier, France. ✉e-mail: kkaneko@shinshu-u.ac.jp

functional groups and some pristine graphene regions is the most stable structure even in liquid water, suggesting the fast dynamic pathways for water transport on the remaining pristine graphene[13]. Willcox and Kim reported that the mixed regions of unoxidized and oxidized graphene sheets play significant role for the rapid water transport across GO membranes[16]. Those remarkable properties of GO for water related techniques can be attributed to its amphiphatic nature and layered structure, yet the microscopic mechanism has not been fully unveiled. In particular, we still do not have enough knowledge on the relationship between the entangle GO layered structure and the uniqueness of water in the interlayer spaces, and the complex structure of GO makes it difficult to compare theoretical and experimental studies.

Here, we propose a realistic layered structure with not only partially staggered, but also wrinkled GO layers based on detailed structural analyses from MD simulations and high-resolution transmission electron microscope (TEM) observation. Furthermore, in-situ X-ray diffraction (XRD) measurements and ex-situ nuclear magnetic resonance (NMR) measurements of water adsorbed on GO demonstrate the coexistence of the two distinct types of adsorbed water molecules in the staggered stacking structure. Our findings provide not only understanding of the anomalous water behavior in the flexible GO frameworks, but also microscopic insight into the higher affinity for $H_2O$ than $D_2O$ near interfaces, which has an impact for example in biological tissues and the human body.

## Results

### MD simulations for the staggered structure of GO

First of all, we performed the MD simulations for the stacked GO structures on water adsorption with reactive force fields to elucidate dynamic GO frameworks. Figure 1a shows a model of simply stacked ultra-large GO sheets, which is the generally accepted GO structure, with and without adsorbed $H_2O$ molecules from MD simulations (Methods). Although the interlayer spacing of the simply stacked ultra-large GO model increases with water adsorption owing to the typical swelling nature of GO, the interlayer spacings are 0.1 and 0.3 nm smaller than the experimental ones identified via XRD with and without water adsorption, respectively (Table 1 and see below).

We also studied the effects of functional groups via the MD simulation of the reduced graphene oxide (rGO) model, developed by removing oxygen functional groups from the ultra-large GO structure (Methods). The simulated sheet structures of the ultra-large GO and rGO layer on MD simulations after equilibrium are shown in Fig. S4a and the layered structure in Fig. S4b. The average interlayer spacings of the simply stacked ultra-large GO and rGO were 0.60 and 0.33 nm, respectively. The interlayer spacing of rGO is similar to that of graphite. Water molecules can be adsorbed on the simply stacked ultra-large GO interlayer spaces causing the interlayer to swell to 0.75 nm. Conversely, the simply stacked rGO exhibit a robust layer to layer van der Waals interaction, making it difficult for $H_2O$ molecules to exfoliate the interlayer structure due to the hydrophobicity, leading to a minor adsorption of $H_2O$. The observed swelling of GO following $H_2O$ adsorption is attributed to the difference between the oxygen-rich sheet structure and the reduced sheet structure of rGO.

The simply stacked model cannot explain well the observed swelling behavior of GO and there should be pillar-like structures in part to maintain the GO interlayer spacing larger than the interlayer contact. Figure 1b shows the MD simulations for a realistic GO model with partially staggered and wrinkling layer structures (Methods), as can be seen in TEM images (Fig. 2a and shown with red arrow in Fig. 3b, c).

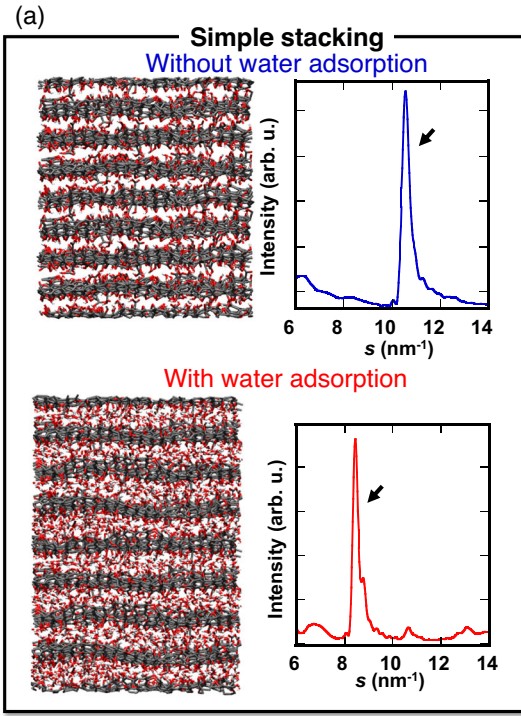
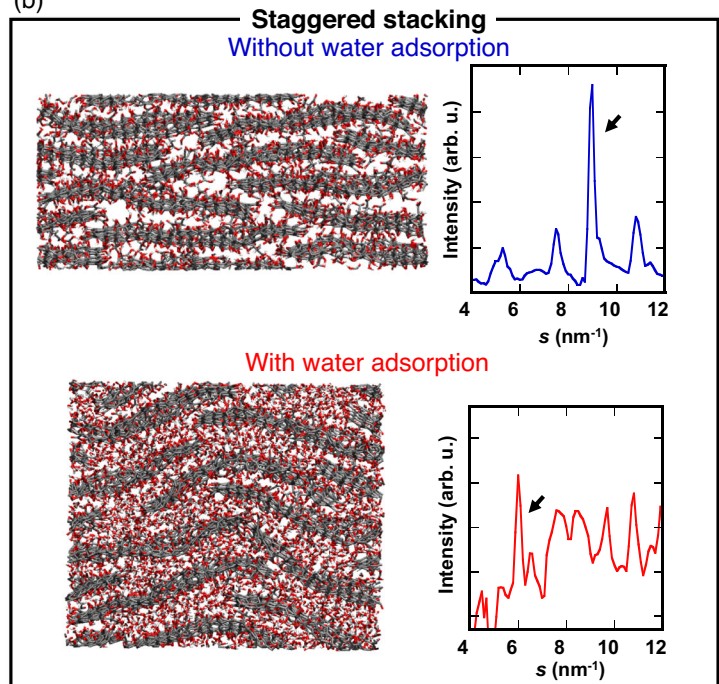

**Fig. 1 | The entangle structures of realistic Graphene oxide (GO). a** Snapshots (left) of the layered structure of simply stacked ultra-large GO from molecular dynamics (MD) simulation with (bottom) and without (upper) $H_2O$ adsorption. Atoms are shown with stick model (gray: carbon, red: oxygen, white: hydrogen). The right figures show the X-ray diffraction (XRD) pattern for the layered structure of GO sheets calculated from the simulated models. The peak pointed by an arrow indicates 001 diffraction of GO layered structure. **b** Snapshots (left) of the partially staggered stacking structure of GO from MD simulation with (bottom) and without (upper) $H_2O$ adsorption, which are created by alternatively stacking of the edge parts of GO sheets and with some compression in the $y$- and $z$-directions. Atoms are shown with stick model (gray: carbon, red: oxygen, white: hydrogen). The right figures show the XRD profile for the layered structure of GO sheets calculated from the simulated models. The peak pointed by an arrow indicates 001 diffraction of GO layered structure.

The simulation results for the staggered stacking structure of GO show the fairly good agreement with the interlayer distances obtained from our in-situ XRD measurements without $H_2O$ adsorption. However, the interlayer distance of the staggered stacking structure of GO with $H_2O$ adsorption (Fig. S12) was 0.2 nm smaller compared with the XRD result at $P/P_0 = 0.8$ (i.e., $d = 0.85$ nm), although the value became improved compared with that of simple stacked ultra-large GO structure simulation. We ascribe this to the flattening of the wrinkling parts of GO layers in the staggered stacking structure with $H_2O$ adsorption, evidenced by the X-ray scattering intensity increase with water adsorption (see below). However, in the realistic GO structure, the wrinkling and the staggered structures cannot be entirely flattened because of the meso- to macroscopic heterogeneities on hydrophilic and hydrophobic parts, resulting in maintaining the metastable wrinkling and staggered stacking structure with $H_2O$ adsorption. It is difficult to reproduce the anisotropy on the $H_2O$ adsorption with molecular simulation perfectly, because of the limitation for the simulation length. In order to adjust the staggered stacking structure of GO with $H_2O$ molecules, we performed additional MD simulations to reduce the length of simulation box $L_x$ form 8.026 nm to ca. 6.4 nm with the pressure in the $y$- and $z$- directions controlled to 0.025 atm. The obtained structure was equilibrated by the *NPT*-MD run of 50 ps at 298 K and 0.025 atm. In the bottom snapshot of Fig. 1b, wrinkling and staggered GO sheets can be seen in the structure with $H_2O$ adsorption, maintaining the interlayer space wider. After the additional *NPT*-MD run for 10 ps, the squared displacements of the randomly elected 85 of $H_2O$ molecules on staggered stacking structure of GO were obtained from the plots of time versus squared displacement (Fig. 2b). The statistical distribution of the squared displacements at 10 ps is shown in Fig. 2c.

In the staggered stacking structure of GO, the interlayer spacing is wider than that in the simply stacked ultra-large GO model, allowing for a good agreement between simulated and experimental XRD results, including upon water adsorption (Table 1).

## Precise analysis of GO structures from high-resolution TEM measurements

Furthermore, we described the staggered stacking structure of realistic GO through the precise analysis of high-resolution TEM images. Figure 3a shows the TEM image of stacked GO layers, highlights the lateral size with yellow lines (more detailed one is shown in Fig. S16), and the lateral size distribution curve (Fig. 3e) indicates that the average GO layer size is approximately 2 nm. Evidently, wrinkling GO layers form the stacked structures, resulting in staggered stacking

**Table 1 | Interlayer spacing between GO layers determined from X-ray diffraction (XRD) measurements and the simulated GO structures**

| | Experimental | Simulated | |
|---|---|---|---|
| | XRD | Simple stacking | Staggered stacking |
| Without water | 0.71 nm | 0.60 nm | 0.70 nm |
| With water | 1.05 nm (@$P/P_0$ = 0.8) | 0.75 nm | 1.05 nm |

Since $P_0$ indicates a saturation vapor pressure of water, $P/P_0$ means relative vapor pressure of water.

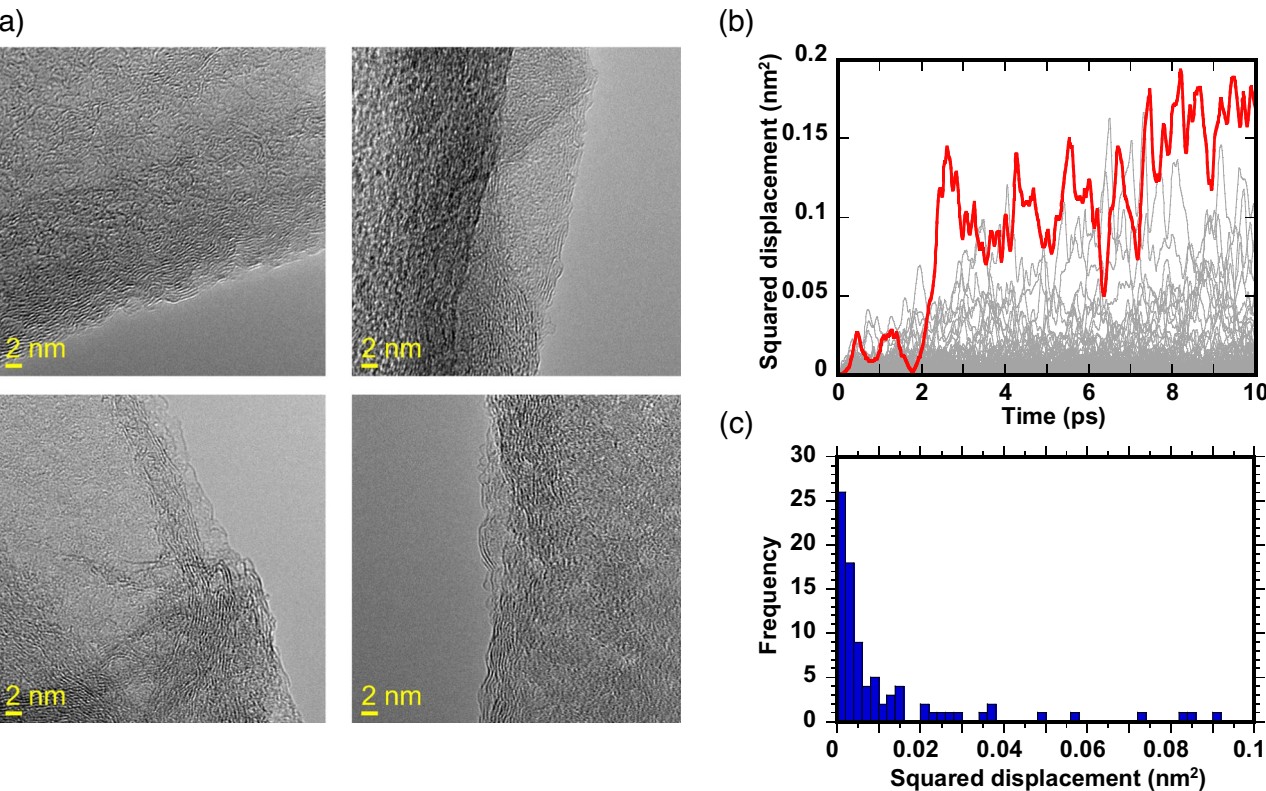

(a)

(b)

(c)

**Fig. 2 | Fast diffusion of water molecules in staggered interlayer spaces of Graphene oxide (GO). a** Transmission electron microscope (TEM) images of layered structure of GO. Yellow scale bar shows the size of 2 nm. **b** Squared displacements of randomly selected 85 molecules of $H_2O$ in the interlayer spaces of partially staggered stacking structure of GO from molecular dynamics simulation. The red line shows the trajectory of the fastest $H_2O$ molecule. **c** The statistical distribution of squared displacements of 85 molecules of $H_2O$ in the interlayer spaces of partially staggered stacking structure of GO at 10 ps. Although 58% of $H_2O$ molecules exhibit slow diffusion (i.e., less than 0.005 nm² per10 ps) − which is similar to the water confined in the interlayer spaces of clay materials[19] − a significant number of $H_2O$ molecules show fast diffusion, even higher than compared with bulk water (0.023 nm² per 10 ps) as shown by the red line in (**b**).

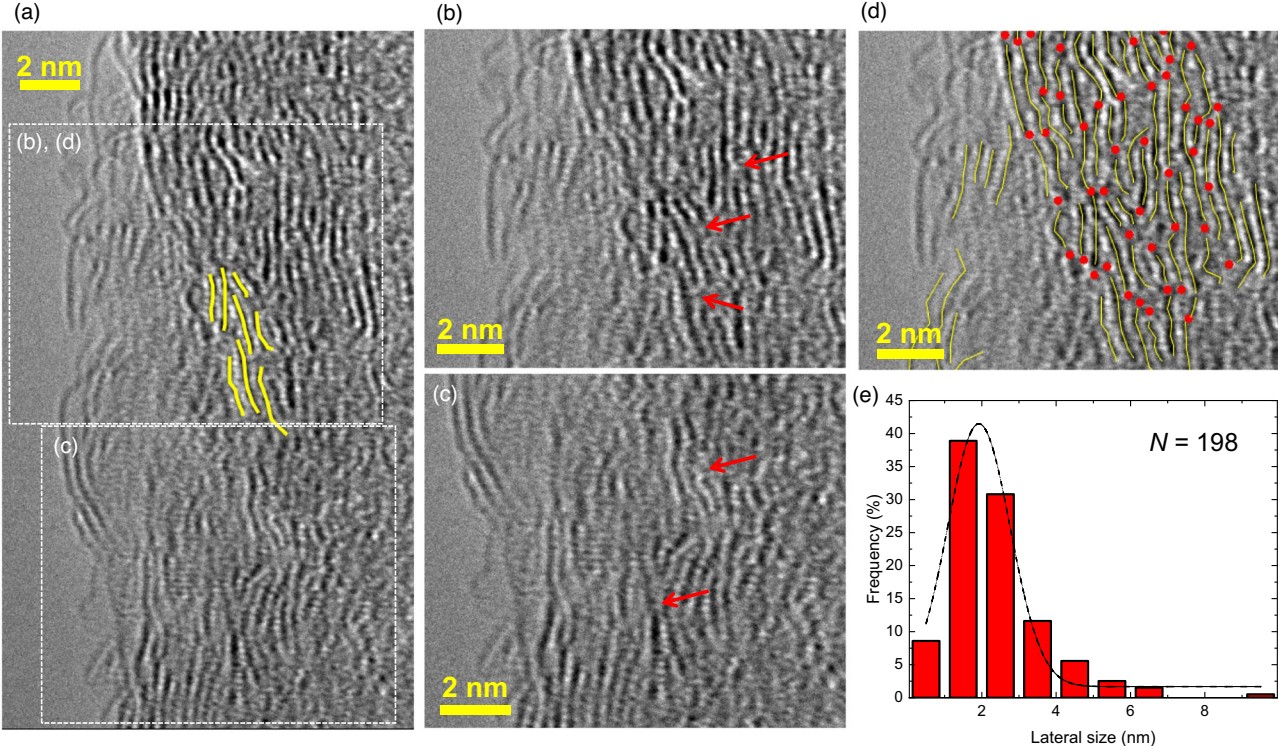

**Fig. 3 | High-resolution Transmission electron microscope (TEM) images of the staggered structures in Graphene oxide (GO). a** The lateral size of GO sheets highlighted with yellow lines in a TEM image. The areas surrounded by white break lines are closed-up in the images of (**b**), (**c**), and (**d**). Figure 3b, d show the same position of the image. **b**, **c** The staggered and relatively widened interlayer spaces of GO as shown by red arrows. **d** The staggered structures in the layers with the number density of 0.8 nm$^{-2}$, as shown by red dots. **a–d** Yellow scale bar shows the size of 2 nm in these images. **e** The lateral size distribution of GO sheet structure with 198 specimens. The break line indicates a result of Gaussian curve fit for the distribution.

structures with relatively widened interlayer spacing, as shown by the red arrows in Fig. 3b, c. Based on statistical analysis of the TEM images, the number density of the staggered and widened interlayer structure is 0.8 nm$^{-2}$ as shown by the red dots in Fig. 3d.

The lateral size of the graphene-like sheet structures of GO was also characterized using the G and D band intensity ratio of the Raman spectrum[18] (i.e., $L_a = 12$ nm) and by the width of 10 and 11 diffraction peaks (i.e., the crystallite sizes of $L_{10} = 11$ nm and $L_{11} = 6.5$ nm) from the XRD measurement as shown in Tables S2 and S3. These results indicate that the lateral size of graphene-like sheets is ca. 10 nm.

The various structure characterizations indicate that the 2-nm GO layer is a primitive unit as observed in the stacking structure determined by TEM measurements, thereby underestimating the lateral size of graphene sheets owing to the limited focus regions by TEM observation. In fact, XRD and Raman spectroscopy showed that the primitive units of GO layers may be sequentially combined to form relatively large units. In our MD simulation for the staggered stacking structure of GO layers, the length of GO sheets in the x-axis was ca. 4.5 nm, which is comparable and intermediate size between the primitive unit size (i.e., 2 nm) and relatively large graphene-like sheet size (i.e., 10 nm). The primitive unit size of GO layers is an essential influencing factor for the structural flexibility on the staggered stacking structure.

The staggered stacking structure not only maintains the interlayer spacing wider but also moves the "free" mobile water molecules in the hydrophobic regions of the interlayer space showing ultrahigh permeances of water molecules reported by ref. 8 for GO membranes. Figure 2b shows the squared displacements of H$_2$O molecules in the staggered stacking GO model, and the statistical distributions are shown in Fig. 2c. Although 58% of H$_2$O molecules exhibit slow diffusion (i.e., less than 0.005 nm$^2$/10 ps)[19], a significant number (15%) of H$_2$O molecules show the fast diffusion even higher than compared with

bulk water (0.023 nm$^2$/10 ps) as shown by the red line in Fig. 2b, coinciding with molecular simulation studies[13,20].

## H$_2$O selective adsorption on GO
Figure 4a shows the single component H$_2$O and D$_2$O vapor adsorption isotherms of GO at 298 K. The water adsorption isotherms are of type IV in IUPAC classification[21], indicating gradual uptake of water molecules on hydrophilic functional groups (i.e., -OH, -C = O etc.) of GO[22–24] from low $P/P_0$. The H$_2$O adsorption capacity of 30 mmol g$^{-1}$ of GO at $P/P_0 = 0.98$ is comparable to that of microporous carbons[21,25–28]. Here $P_0$ indicates the saturation vapor pressure. Interestingly, the adsorption amount of D$_2$O at $P/P_0 = 0.98$ is 23% smaller than that of H$_2$O, showing the D$_2$O-phobicity of GO (Fig. 4d, e, see the details in the next section).

Moreover, the mixed-vapor adsorption of H$_2$O and D$_2$O was conducted using laboratory-designed equipment with a mass spectrometer (see Fig. S2a) to confirm the H$_2$O selective adsorption on GO (see details of the experimental setups in the supplemental information (SI)). Herein, we describe the composition of the mixture with the H/D ratio, as HDO molecules inevitably form in the H$_2$O/D$_2$O mixture. We refer to the hydrogen isotopic water mixture as an "H$_2$O/D$_2$O mixture," even though the mixture contains HDO whose composition varies with the mixture composition of H$_2$O and D$_2$O at a constant temperature.

In this measurement, the vapor of the liquid mixture of H$_2$O:D$_2$O with a mole ratio of 1.13:1.00 (i.e., H/D = 1.13) was adsorbed on GO at 298 K and $P/P_0 = 0.94$ for 24 h. We determined the H/D ratio of the adsorbed H$_2$O/D$_2$O mixture on GO as follows. We fully desorbed the H$_2$O/D$_2$O mixture adsorbed on GO at 333 K for 2 h and collected the desorbed H$_2$O/D$_2$O mixture using a cold trap at 77 K. We confirmed the entire desorption of H$_2$O and D$_2$O molecules from GO using weight loss measurements of GO adsorbing water molecules under the desorption conditions of 333 K in vacuum (<0.1 Pa) for 2 h (Table S4). Furthermore, we conducted thermogravimetric (TG) measurements

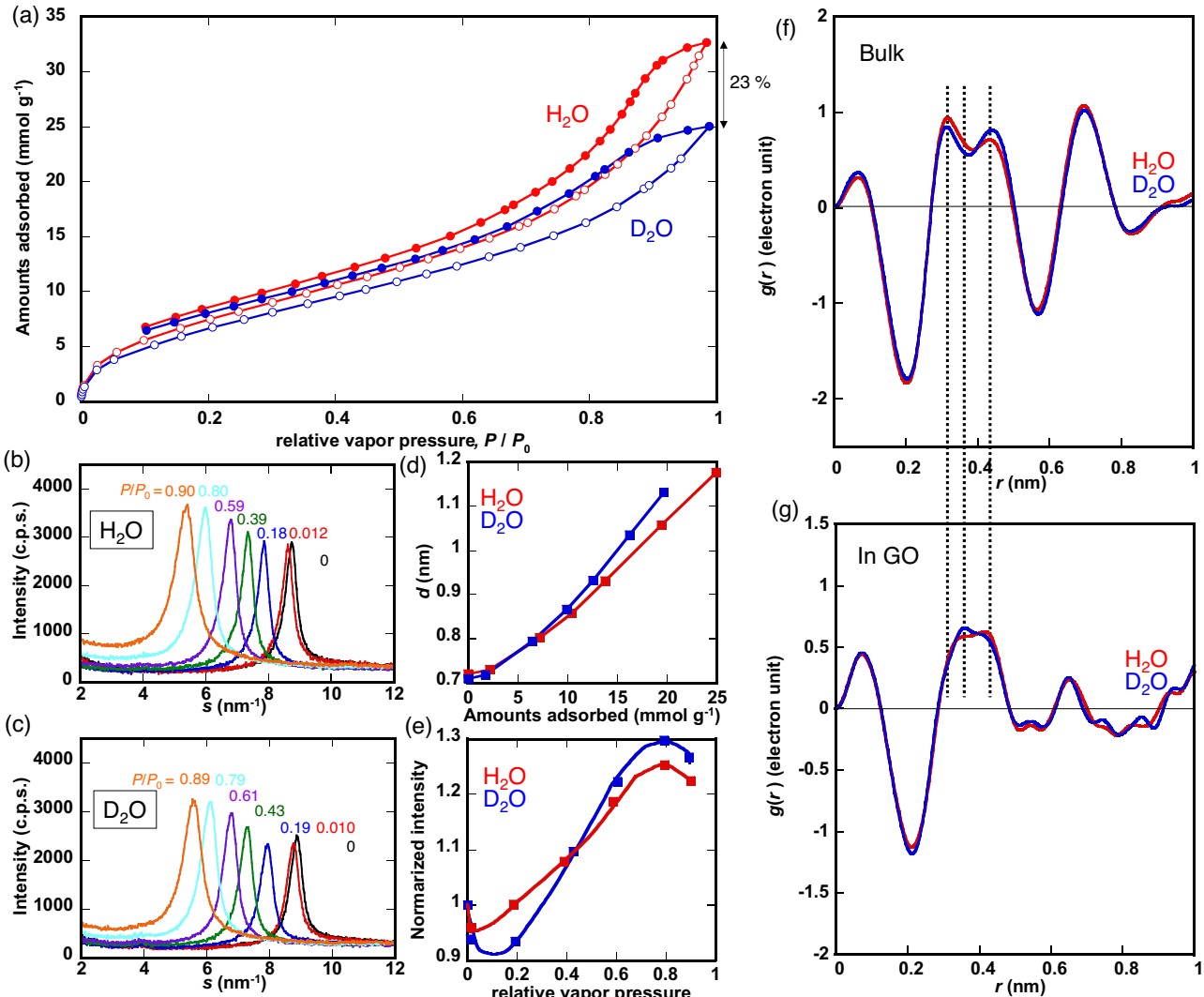

**Fig. 4 | Preferential adsorption of H₂O rather than D₂O on Graphene oxide (GO).** **a** Single component adsorption isotherms of $H_2O$ (red) and $D_2O$ (blue) vapor on GO at 298 K. Adsorption and desorption branches are shown with the open and closed circles, respectively. Water uptakes (in units of mmol g⁻¹) are plotted against relative vapor pressures ($P/P_0$). Here $P_0$ indicates the saturation vapor pressure. The water adsorption isotherms are of type IV in IUPAC classification[21], indicating gradual uptake of water molecules on hydrophilic functional groups (i.e., -OH, -C = O etc.) of GO[22–24] from low $P/P_0$. **b** Diffraction peaks of 001 plane of GO adsorbing H₂O at $P/P_0$ = 0(black), 0.012(red), 0.18(blue), 0.39(green), 0.59(purple), 0.80(sky blue) and 0.90(orange) by in-situ X-ray diffraction (XRD) measurement. **c** Diffraction

peaks of 001 plane of GO adsorbing D₂O at $P/P_0$ = 0(black), 0.010(red), 0.19(blue), 0.43(green), 0.61(purple), 0.79(sky blue) and 0.89(orange) by in-situ XRD measurement. **d** GO interlayer spacing (as evaluated by XRD) as a function of water uptakes (H₂O: red dots and a line, D₂O: blue dots and a line). **e** Pressure ($P/P_0$) dependence of the 001 diffraction peak intensity upon water adsorption (H₂O: red dots and a line, D₂O: blue dots and a line). The intensities are normalized by that of GO under a vacuum (i.e., $P/P_0$ = 0). **f** Electron radial distribution functions ($g(r)$) of bulk H₂O (red) and D₂O (blue) obtained by Fourier transformation of the X-ray structure functions. **g** Electron radial distribution functions of H₂O (red) and D₂O (blue) adsorbed on GO at $P/P_0$ = 0.9.

and differential thermal analysis (DTA) of GO adsorbing water (see Figs. S14 and S15), and we confirmed there is no clear difference in the desorption temperatures of physisorbed H₂O and D₂O.

Mass spectral measurements of $I^{18}$ and $I^{20}$, which are the mass intensities at $m/z$ = 18 and 20, for the desorbed vapor were then conducted to determine the H/D ratio of the H₂O/D₂O mixture.

Figure 2S(b) shows the time courses of the intensity-ratios ($I^{18}/I^{20}$) for the feed vapor (red) and the desorbed vapor after adsorption on GO (blue). The stationary $I^{18}/I^{20}$ values of the feed vapor ($I^{18}/I^{20}$ = 1.25) and the desorbed mixture ($I^{18}/I^{20}$ = 2.40) were determined by averaging the curves over 100 s from 200 s to 300 s to obtain the reliable $I^{18}/I^{20}$ values of the equilibrated composition of the vapors due to the faster evaporation rate of lighter isotopes of H₂O in the early stage of the measurements. The preferential adsorption of H₂O on GO over that of D₂O was confirmed by the higher mass intensity ratio of $I^{18}/I^{20}$ for the desorbed vapor than that for the feed vapor. The H/D ratio of the H₂O/

D₂O mixture was determined from a calibration curve of the mass spectral intensity ratio of $I^{18}/I^{20}$ versus the H/D ratio of the H₂O/D₂O mixture (Fig. S13). The corresponding H/D ratios of the feed vapor and desorbed vapor after adsorption on GO were 1.13 and 1.70, respectively. This indicated a higher H content in the adsorbed mixture on GO than in the feed vapor (i.e., the amount of H₂O is greater than that of D₂O in the adsorbed mixture).

Here, an unclear comparison between single component adsorption and mixed vapor adsorption should be avoided due to the predominantly formed HDO in the H₂O/D₂O mixture, as chemical exchange occurs based on the following equilibrium constant:

$$K = \frac{[\text{HDO}]^2}{[\text{H}_2\text{O}][\text{D}_2\text{O}]}$$
$$= 3.85 (\text{at 298K})$$

Therefore, the determined H/D ratio of hydrogen isotopic water mixtures adsorbed on GO was compared with the H/D value obtained from single component adsorption measurements as follows.

The adsorption amount of the hydrogen isotopic water mixture on GO was 399 mg g$^{-1}$, as determined by the weight measurement following the mixed $H_2O$/$D_2O$ vapor adsorption. The corresponding adsorption amounts of $H_2O$, $D_2O$, and HDO in the mixed-vapor adsorption were 8.36, 2.91, and 9.68 mmol g$^{-1}$, respectively. Here, H/D = 1.70 and the equilibrium constant ($K$ = 3.85) for the isotopic exchange reaction were used to determine the adsorption amounts for each component. The adsorption amounts for single-component adsorption at $P/P_0$ = 0.94 were 14.1 and 10.9 mmol g$^{-1}$ for $H_2O$ and $D_2O$, respectively, which were scaled by 0.5 to compare with those of the 1:1 mixed-vapor adsorption. The corresponding H/D ratio of single-component adsorption was 1.29, and the higher H/D ratio under mixed-vapor adsorption (i.e., H/D = 1.70, see Table S1). This indicates that the $D_2O$-phobicity of GO was promoted because of selective $H_2O$ adsorption even on the unfavorable adsorption sites of the oxygen functional groups for $D_2O$ as compared with that of single-component adsorption.

These results are inconsistent with those of the pervaporation membrane separation reported by ref. [10] The contradiction between our results and those of the cited references are attributed to the difference in adsorption separation (for this study) and membrane separation (in refs. [10,11]). The pervaporation separation using membrane filters is related to numerous factors such as gas permeation or adsorption to the solid, the gas diffusion in the solids owing to the difference in gas pressure or concentration, and the gas desorption from the solid. Conversely, the adsorption separation includes only one of these steps and the most important factor is the interaction between gas molecules and the solids. Furthermore, pervaporation is conducted based on high temperature (i.e., ca.373 K in ref. [10]) to permeate vapor molecules to solid more rapidly even when the adsorption separation of $H_2O$/$D_2O$ was conducted at 298 K. Generally, the quantum effects between $H_2O$ and $D_2O$ are significant at ambient temperature and vanish with increasing temperature. At this point, $D_2O$ selective filtration by pervaporation may occur primarily because of the kinetic mechanism underlying heavier molecules of $D_2O$ and not because of quantum effects.

The evident $D_2O$-phobicity of GO is understandable from quantum simulation: adsorption of $D_2O$ having a stronger hydrogen bonding between $D_2O$ molecules occurs at only the optimum arrangement of surface functional groups, whereas $H_2O$ of the weaker hydrogen bonding nature can adsorb at any surface functional group sites (see Fig. S1 and SI).

### In-situ XRD measurements of GO adsorbing water

Figure 4b, c show in-situ XRD profiles of GO as a function of $P/P_0$, which shows a difference depending on the $H_2O$ and $D_2O$ nature. Here, $s$ (= $4\pi\sin\theta/\lambda$) is the scattering parameter. The 001 diffraction peak of GO at $s$ = 8.7 nm$^{-1}$ shifts to a lower angle with increasing $P/P_0$, indicating the evident swelling of GO interlayer spaces, similar to montmorillonite[29], because of the flexible GO interlayer frameworks[4,5,14]. Correspondingly, the interlayer spacing and diffraction intensity vary depending on adsorption of $H_2O$ and $D_2O$. The increase of the interlayer spacing on $D_2O$ adsorption is more marked than on $H_2O$ adsorption with small amounts of adsorbed $D_2O$ due to the above-mentioned $D_2O$-phobicity of GO (Fig. 4d). The scattered intensity vs $P/P_0$ curves have an initial drop and a maximum near $P/P_0$ = 0.8 (Fig. 4e). The drop and maximum for $D_2O$ are more remarkable than those for $H_2O$.

The $D_2O$-phobicity of GO should distort the local interlayer structures, which decreases the scattering intensity at the initial stage of $D_2O$ adsorption, giving the scattering intensity drop more efficiently from $H_2O$-adsorbed GO stacking layers. The subsequent water adsorption leads to the intensity increase because of the higher electron density contrast between the water-adsorbed GO layer and the

interlayer space after the initial intensity drop. The sufficiently adsorbed water molecules fill the interlayer spaces near $P/P_0$ = 0.8, resulting in the observed intensity reduction. Then, $D_2O$ with a smaller adsorption amount provides similar intensity changes as $H_2O$ with a larger adsorbed quantity, indicating the larger population of non-hydrogen bonded $D_2O$ molecules in the interlayer spaces.

Figure 4f, g show the electron radial distribution functions (ERDFs) of $H_2O$ and $D_2O$ in bulk and adsorbed on GO at $P/P_0$ = 0.9, obtained from the XRD measurements. The intensity near 0.3–0.4 nm for $H_2O$ is larger than that for $D_2O$ in bulk because bulk $H_2O$ has more non-hydrogen bonded-water than bulk $D_2O$ that exhibits stronger hydrogen bonding of $D_2O$[30]. However, very interestingly, the opposite tendency was observed for $H_2O$ and $D_2O$ adsorbed on GO; the peak intensity of $D_2O$ at 0.35 nm is larger than that of $H_2O$ on GO, showing more non-hydrogen bonded $D_2O$ than $H_2O$ on GO. This also stems from $D_2O$-phobicity of GO, and the observed difference of the adsorbed states between $H_2O$ and $D_2O$ on GO is discussed with the relevance to unique staggered stacking structure, later.

### Ex-situ NMR measurements of GO adsorbing water

Figure 5a, b show ex-situ $^1$H-NMR spectra of GO with adsorbed $H_2O$ and $D_2O$ at $P/P_0$ = 0.9, respectively. The $^1$H-NMR spectrum of GO with adsorbed $H_2O$ shows a broad and asymmetric profile, which is characteristic for the immobile $H_2O$ in the anisotropic environments of GO interlayer spaces[31]. The adsorbed $H_2O$ gives a chemical shift larger than that of bulk $H_2O$ (i.e., 4.8 ppm from tetramethylsilane, TMS), indicating the presence of hydrogen bonds stronger than those in bulk $H_2O$. Importantly, $H_2O$ adsorbed on GO shows an evident deshielding effect (i.e., higher chemical shifts) suggesting the hydrogen bonding with surface functional groups of GO, being completely different from $H_2O$ adsorbed on microporous carbons[32].

The broad asymmetric peak of GO with adsorbed $H_2O$ is deconvoluted into two Gaussian peaks with peak positions at 8.67 ppm and 6.92 ppm, whose populations are 74% and 26%, respectively (Table 2). On the other hand, the peak deconvolution of GO with adsorbed $D_2O$ gives two peaks at 8.36 ppm and 6.83 ppm whose populations are 65% and 35%, respectively (Table 2). Since deshielding can occur due to a stronger hydrogen bonding network with water molecules or functional groups, the number of strongly hydrogen bonded $D_2O$ molecule is less compared to that of $H_2O$ on GO. The ex-situ NMR study support the $D_2O$-phobicity obtained from the above ERDF analyses.

The wide line $^2$H-NMR study supports the above discussion from a dynamic point of view, although it can only provide the dynamics of $D_2O$ having the magnetic quadrupole moment. Figure 6a shows the temperature dependence of the wide line $^2$H-NMR spectra of $D_2O$ adsorbed on GO near $P/P_0$ = 1. A sharp peak appears in the broad Gaussian shape on lowering the temperature from 293 K to 253 K. Widening of the line shape in the low temperature range of 173 K to 153 K is ascribed to the slowdown of molecular motions of all adsorbed $D_2O$ molecules. The new sharp peak in the broad one at 253 K is characteristic for $D_2O$ on GO, suggesting the coexistence of kinetically different water molecules in the staggered stacking structure of GO, and we assign it to slow chemical exchange of the two kinds of adsorbed water molecules with different mobilities as found from MD simulations. The two kinds of adsorbed $D_2O$ are dynamically independent of each other at 253 K, with low chemical exchange rate. However, the exchange rate increases with elevation of the temperature, resulting in the peak broadening. The two kinds of adsorbed $D_2O$ having completely different mobilities should be associated with two adsorbed states of water on GO: The highly mobile water in the hydrophobic regions of the interlayer space without hydrogen bonds and the slow water strongly hydrogen bonded with hydrophilic functional groups or other water molecules.

In this work, we firstly showed an important aspect of GO on water adsorption from MD simulations. Although the flexible nature of the

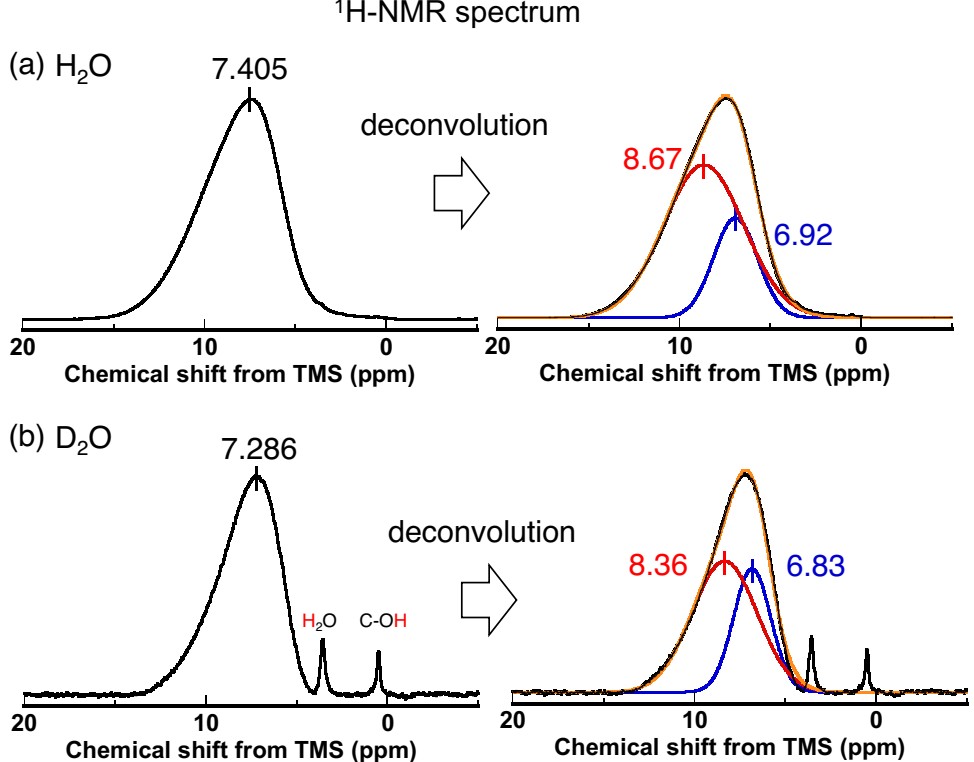

**Fig. 5 | Lower population of strongly hydrogen bonded $D_2O$ compared with that of strongly hydrogen bonded $H_2O$ on Graphene oxide (GO), evidenced by ex-situ $^1H$ nuclear magnetic resonance (NMR) measurements. a** Ex-situ $^1H$-NMR spectrum for GO adsorbing $H_2O$ at relative pressure $P/P_0 = 0.9$ (left) and the spectrum deconvoluted with two Gaussian functions (red and blue curves in right Fig.). Here $P_0$ indicates the saturation vapor pressure. The yellow curve shows the sum of the two Gaussian functions. The abscissa is shown with chemical shifts from that of tetramethylsilane (TMS). The $^1H$-NMR spectrum of GO with adsorbed $H_2O$ shows a broad and asymmetric profile, which is characteristic for the immobile $H_2O$ in the anisotropic environments of GO interlayer spaces[31]. **b** Ex-situ $^1H$-NMR spectra for GO adsorbing $D_2O$ at $P/P_0 = 0.9$ (left) and the spectrum deconvoluted with two Gaussian functions (red and blue curves in right Fig.). The yellow curve shows the sum of the two Gaussian functions. The abscissa is shown with chemical shifts from that of tetramethylsilane (TMS).

interlayer spacing of GO could be attributed to its oxygen-rich sheet structure, which was not observed with rGO, the simple stacking model of ultra-large GO sheets, which is a generally accepted GO model, cannot adequately explain the swelling behavior of GO observed in the XRD experiments. Our MD simulations for the staggered stacking of flexible GO sheets can effectively explain the swelling nature of GO upon water adsorption, and TEM observation also supports the existence of the non-regular staggered stacking structure of GO. Furthermore, our XRD and NMR results for the unique adsorbed water behaviors agree with the MD simulation for the staggered stacking structure of GO from both kinetic and structural aspects.

## Discussion

Here, we discuss the specific $H_2O/D_2O$ recognition function of GO due to $D_2O$-phobicity. The above results indicate that the interlayer space of staggered stacking structure of GO with oxygen functional groups and the interlayer distances efficiently distinguish the $H_2O/D_2O$ through their hydrogen bonding differences, being similar to biomolecules in the human body[33].

## Table 2 | Chemical shits and populations of the deconvoluted peaks in $^1H$-NMR spectra for GO adsorbing $H_2O$ and $D_2O$

|  | $H_2O$ | | $D_2O$ | |
| --- | --- | --- | --- | --- |
|  | Chemical shift (ppm) | Population of peak area (%) | Chemical shift (ppm) | Population of peak area (%) |
| Peak 1 | 8.67 | 74 | 8.36 | 65 |
| Peak 2 | 6.92 | 26 | 6.83 | 35 |

The difference between the adsorption amounts of $H_2O$ and $D_2O$ molecules on GO increases as adsorption progresses, but non-negligible preferential adsorption of $H_2O$ is observed in the initial stage of single component adsorption isotherms of $H_2O$ and $D_2O$ on GO at very low $P/P_0 < 0.03$ as shown in the magnified figure of Fig. S6. The adsorption amount of $H_2O$ exceeds that of $D_2O$ at all pressure levels, including this very low-pressure region and the difference is equivalent to 12% of the adsorption amounts at $P/P_0 = 0.024$, indicating the clear $D_2O$-phobicity of GO functional groups. This is not only explained by the quantum effects of $H_2O/D_2O$ but also by the staggered stacking structure of GO.

We propose a microscopic adsorption mechanism (Fig. 6b). Predominant water molecules are two-fold coordinated hydrogen atoms bonded with surface functional groups on the staggered stacking structure of GO. $H_2O$ molecules exhibit more optimized interaction with the surface functional groups on GO through their excellent hopping motion. Furthermore, $H_2O$ molecules can more easily form a dangling hydrogen bond between functional groups than $D_2O$ molecules owing to the enhanced localization of heavy D atoms[34]. As a result, the $D_2O$ molecule is less adaptable to the strongly constrained local arrangement of surface functional groups on the staggered stacking structure of GO, giving the observed $D_2O$-phobicity of a smaller adsorbed amount than $H_2O$ even at low $P/P_0$. Here, the staggered stacking structure of GO plays a very important role for the $H_2O/D_2O$ recognition by keeping the interlayer spacing wider by 0.1–0.3 nm than that without the staggered interlayer structure, resulting in a monolayer of adsorbed water on GO without the formation of highly developed hydrogen bonding networks between water molecules as bulk water.

(a)            ²H-wideline NMR              (b)

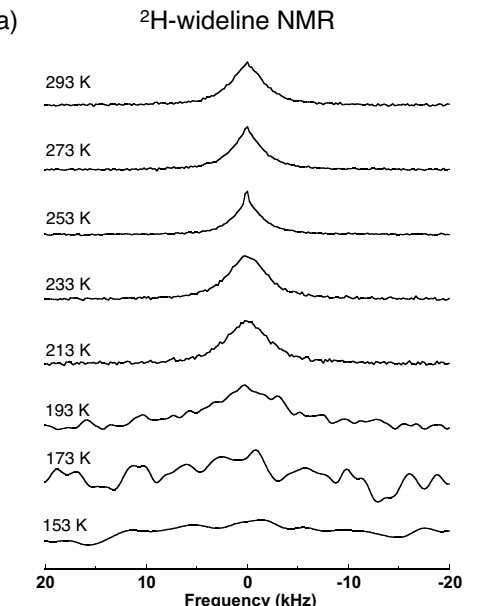

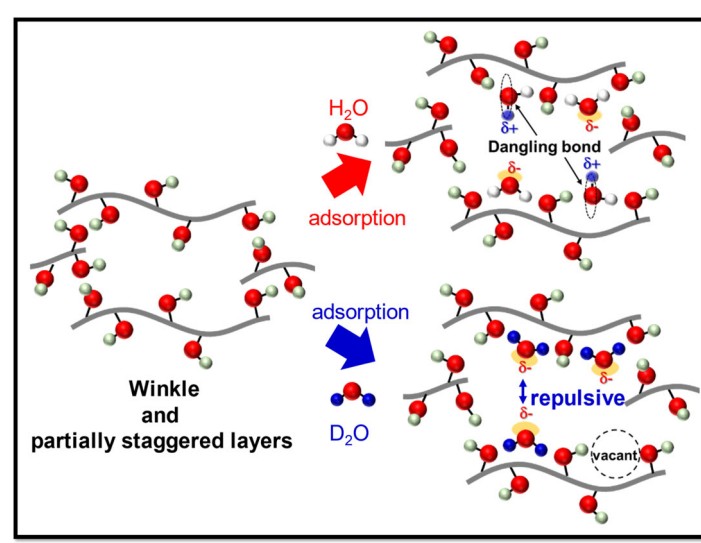

δ+, δ-: Partial charges

**Fig. 6 | Coexistence of kinetically different water molecules in the interlayer spaces of partially staggered stacking structure of Graphene oxide (GO), evidenced by ²H nuclear magmatic resonance (NMR) measurement. a** Temperature dependence of wide line ²H-NMR spectra for $D_2O$ adsorbed on GO near relative pressure $P/P_0 = 1$. Here $P_0$ indicates the saturation vapor pressure. As temperature is lowered from 293 K to 253 K, a sharp peak appears in the broad Gaussian shape. The new sharp peak in the broad one at 253 K is characteristic for $D_2O$ on GO, suggesting the coexistence of kinetically different $D_2O$ molecules on GO, and we assign it to slow chemical exchange of the two kinds of adsorbed $D_2O$ molecules with different mobilities. Widening of the line shape in the low temperature range of 173 K to 153 K is ascribed to the slowdown of molecular motions of all adsorbed $D_2O$ molecules. **b** Plausible adsorption models of $H_2O$ and $D_2O$ in the interlayer spaces of realistic staggered stacking structure of GO (left): $D_2O$ molecules are less adaptable to the strongly constrained local arrangement of surface functional groups on GO (bottom of right), giving a lower adsorbed amount than $H_2O$ (top of right). Moreover, in the adsorbed $D_2O$ molecules on GO layers, the O atoms tends to point to the center of the interlayer spaces resulting in the repulsive interaction between their permanent dipole moments of $D_2O$. The total polarization (i.e., the Coulombic repulsions, consequently) of each GO layer adsorbing $D_2O$ molecules is larger than that of $H_2O$ due to the localized $D_2O$ molecules on GO even with small amounts adsorbed, resulting in the larger interlayer spacing with the smaller amount of adsorbed. Here, the staggered stacking structure of GO plays a very important role for the $H_2O/D_2O$ recognition by keeping the interlayer spacing wider by 0.1–0.3 nm than that without staggered interlayer structure, resulting in a monolayer of adsorbed water on GO without the formation of highly developed hydrogen bonding networks between water molecules. δ+ and δ- represent the partial charges on water molecules.

For the practical use of GO as $H_2O/D_2O$ separation filters, the presence of HDO is inevitable owing to proton exchange. Furthermore, the equilibrium constant $K$ could differ from that of the bulk value because the interactions between the hydrogen isotopic water molecules and substances would differ.

Moreover, the decompositions of the functional group are very important for molecular separation with GOs. The ab initio MD simulation by Mouhat et al. showed that the decomposition of GO functional groups was seldom observed and most of the system was unreactive[13]. In our MD simulation, we observed certain decompositions of hydroxyl functional groups and the hydrogen transfer from a physisorbed $H_2O$ to a hydroxyl functional group; however, these are not major events as shown in Fig. S5.

In this study, we do not treat $D_2O$ molecules with our reactive MD simulation because our force field does not include deuterium. Recently, Zhang et al. proposed a reactive force field that can adequately characterize the differences in the radial distribution function (RDF), self-diffusion constant, and vibrational spectrum between heavy and light water and is suitable for elucidating the $H_2O/D_2O$ differences in the GO structure with MD simulation[11,35].

Although we simulated the microscopic behaviors of $H_2O$ molecules in GO interlayer spaces based on MD with reactive force field, the observed differences between the fluidic behaviors of $H_2O$ and $D_2O$ can be elucidated more clearly from the quantum effects with the aid of path integral simulations like Centroid MD[36,37]. A detailed study should be conducted with path integral quantum MD simulation with the realistic staggered stacking structure of GO as in our near future works. The knowledge on the detailed mechanism of the higher affinity

of $H_2O$ than $D_2O$ for GO can offer not only new applications of $H_2O/D_2O$ separation with GO membrane but also microscopic insights into the higher affinity for $H_2O$ than $D_2O$ near interfaces, which has an impact for example in biological tissues and the human body[33].

## Methods
### Preparation of GO
GO material was prepared by improved Hummers methods[6,38] with natural graphite of Bay carbon graphite from Michigan. The graphite powder (5 g) was inserted in the mixture of 200 mL $H_2SO_4$ (96%, Wako Chemical Co. Ltd.) and 25 mL $H_3PO_4$ (85% Wako Chemical Co. Ltd.) followed by addition of 25 g $KMnO_4$ powder (Wako Chemical Co. Ltd.). The mixing of the chemicals with exothermic reactions was carried out very slowly under cooling with ice-water bath to control the temperature at $311 \pm 2$ K stirring at 200 rpm for 2 h. Then, the 500 mL of distilled water was added slowly into the mixture followed by addition of 100 mL $H_2O_2$ solution (10%, Wako Chemical Co. Ltd.). Then, the supernatant by centrifugation was collected and was washed with 1 M HCl (Wako Chemical Co. Ltd.) at 3 times and then washed with distilled water to reach the pH of the washed liquid to 7. Finally, we obtained GO powders by freeze-drying of the GO suspension with liquid nitrogen.

### Characterization of GO
The thermogravimetric analysis was conducted with a TG-DTA equipment (Rigaku Co.) under $N_2$ flow at 100 mL min⁻¹. The heating rate was 3 K min⁻¹. The TG result for GO sample shows the large amounts of weight loss at 333 K, indicating desorption of water molecules (Fig. S7). As further heating treatments reduce the oxygen

functional groups of GO, we pre-evacuated GO materials at 333 K for 2 h before the adsorption isotherm measurement, ex-situ NMR, and in-situ XRD measurement in this work.

The high-resolution transmission electron micrographs were obtained using a high-resolution transmission electron microscopy (HR-TEM; JEOL, TEM-2100F, 80 kV). GO was vacuum-dried at 353 K for 2 h prior to the TEM observation. The dried GO powder was lightly pressed on a 150 mesh Cu microgrid with carbons (Okenshouji Co., Ltd.) for the TEM observation; the solvent free method is preferable to avoid the morphological change of GO particles upon wetting and/or dispersion.

$N_2$ adsorption isotherm of GO was measured at 77 K with the volumetric equipment of Autsorb iQ2 (Anton Paar GmbH) (Fig. S8). Single component vapor adsorption isotherms of $H_2O$ and $D_2O$ on GO at 298 K were measured with a volumetric vapor adsorption apparatus of Vstar (Anton Paal GmbH). For the water adsorption and following in-situ XRD measurements, distilled water and deuterium oxide (99.9 atom% deuterated, Sigma-Aldrich co.) were used for $H_2O$ and $D_2O$, respectively. The specific surface areas of GO evaluated from $H_2O$ adsorption isotherm at 298 K and $N_2$ adsorption isotherm at 77 K are 510 $m^2\,g^{-1}$ and 8 $m^2\,g^{-1}$, respectively, indicating the high accessibility of $H_2O$ molecules in the GO interlayer spaces at room temperature.

The Raman spectrum of GO was measured with a Raman spectrometer (NRS-3100, JASCO Co.) using a 532 nm-wavelength laser at a room temperature (Fig. S3b).

## In-situ XRD measurement of GO adsorbing water

For the in-situ X-ray diffraction measurements of GO adsorbing $H_2O$ and $D_2O$ vapors, GO sample was inserted in an XRD measurement chamber connected to a cryostat and a volumetric adsorption line which can control the temperature and vapor pressures. The GO sample was packed into a 1 mm thick slit-shaped cell and the cell was placed in the chamber located at the X-ray irradiation center. The small-angle XRD measurements of GO were performed at several relative pressures of $H_2O$ and $D_2O$ with CuKα X-ray (40 kV, 30 mA) by transmission method using an angle-dispersion diffractometer (Ultima III, Rigaku Co.) at 303 K. The wide-angle XRD measurement of GO adsorbing $H_2O$ and $D_2O$ at $P/P_0 = 0.9$ were performed to obtain the ERDFs using MoKα X-ray (50 kV, 30 mA) by transmission method with the same equipment at 303 K. The wide-angle XRD profile of GO under a vacuum is shown in Fig. S3a. The X-ray scattering measurements for the bulk $H_2O$ and $D_2O$ were conducted with Rapid II (Rigaku Co. MoKα, 50 kV, 30 mA) equipped with a 2D imaging plate as a detector. The evaluation method of ERDFs can be seen as follows[39,40].

The adsorption system can be regarded as a three-phases mixing system constituted with adsorbed molecules (admolecule), solids, and vacant pore spaces. Here, the admolecule is water. The experimental X-ray scattering intensity ($I_{obs}$) of an adsorption system consists of the sum of the self-scattering terms of solid ($I_{sc}^s$) and adsorbed molecules ($I_{sc}^a$), small angle X-ray scattering due to the presence of pores ($I_{saxs}$), and the interference terms between admolecules ($I_{if}^{a-a}$), solid atoms ($I_{if}^{s-s}$), and admolecules and solid atoms ($I_{if}^{s-a}$) with multiplied by several correction factors. Therefore, the $I_{obs}$ is given by the following Eq. 1:

$$I_{obs} = kPGA\{I_{sc}^s + I_{sc}^a + I_{if}^{s-s} + I_{if}^{a-a} + I_{if}^{s-a} + I_{SAXS}\} \quad (1)$$

where $k$ is the coefficient converting the experimental intensity from e. u. (electron unit) to c. p. s. unit. $P$, $G$, and $A$ are the correction factors concerning with polarization, X-ray irradiating volume, and X-ray absorption, respectively. These factors are given by the following Eqs. 2, 3 and 4:

$$P = \frac{1 + \cos^2 2\theta}{2} \quad (2)$$

this is the correction for monochromatization with Kβ filters.

$$G = \frac{1}{\cos \theta} \quad (3)$$

where the cell shape is assumed to be perfectly flat.

$$A(\theta) = \exp(-\mu l / \cos \theta) \quad (4)$$

where $\mu$ is linear absorption coefficient of the system and $l$ is the cell length. However, it is difficult to measure $\mu$ and $l$ experimentally so that we determine $\mu l$ from the following Eq. 4':

$$\log(I^t / I_0^t) = -\mu l \quad (4')$$

where $I^t$ is the transmission intensity when the sample is present, $I_0^t$ is the X-ray transmission intensity when no sample is present. The correction factors, $P$, $G$, and $A$ depend on experimental setup and here we used these for transmission method on slit shaped sample cell with monochromatized MoKα X-ray by a NiKβ filter. The reduced structure function of adsorbed system $S_{ad}(s)$, which is the direct structural information from X-ray scattering, consists of summation of $I_{if}^{a-a}$ and $I_{if}^{s-a}$, divided by sum of squares of the electron numbers on all atoms in the admolecule for the system as shown by the following Eq. 5:

$$S_{ad}(s) = \frac{I_{if}^{a-a} + I_{if}^{s-a}}{\sum_i^n Z_i^2} \quad (5)$$

where $Z_i$ is the electron number of atom $i$ in the admolecule. To obtain $S_{ad}(s)$, at first, the $I_{saxs}$ was subtracted using power law of intensity to $s$ at small angle region (i.e., $I_{saxs} \sim s^{-b}$, here $b$ is a positive constant). Both of $I_{sc}^s$ and $I_{if}^{s-s}$ were obtained from the scattering intensity $I_{obs}^s$ of GO itself in a vacuum under an assumption of no structural change of GO by water adsorption. The $I_{sc}^a$ of water is defined as following Eq. 6:

$$I_{sc}^a = \sum_m^n f_m^2 + \sum_m^n i_m^{inc} \quad (6)$$

where $f_m$ and $i_m^{inc}$ are the atom scattering factor and the incoherent scattering intensity of $m$th atom in the admolecule, respectively.

The electron radial distribution function (ERDF, $g(r)$) of the adsorbed system can be driven from the $S_{ad}$ by Fourier transformation as following Eq. 7:

$$g(r) = 4\pi r^2 (\rho - \rho_0) = \frac{2r}{\pi} \int_0^{s_{max}} s S_{ad}(s) \exp(-Bs^2) \sin sr \, ds \quad (7)$$

where $\rho$ and $\rho_0$ are the local and average electron densities at a distance $r$ for the system, respectively. Here, $\exp(-Bs^2)$ is a convergence factor and the value of 0.02 Å was used for $B$.

## Ex-situ NMR measurement of GO adsorbing water

The ex-situ NMR measurements of GO adsorbing water were conducted using ECA-500 NMR spectrometer (JEOL Ltd.) equipped by a superconducting magnet with an external magnetic field of $B_0 = 11.747$ T, which gives the Larmor frequencies of 500.16 MHz for $^1H$ nuclei and of 76.78 MHz for $^2H$ nuclei, respectively. Water adsorbed GOs were prepared as followed: The pretreated GOs were exposed under the saturated water vapor at a room temperature in NMR glass tubes, the uptakes of water were controlled until reaching the appropriate adsorption amount by changing adsorption time, and then the glass tubes were sealed with flame, taking care of desorption of water by heating. The resultant samples are expected to have objective adsorption amounts ($w$) of $H_2O$ (i.e., $w = 57$ mg $g^{-1}$ and 520 mg $g^{-1}$) and $D_2O$ (i.e., 320 mg $g^{-1}$) on GOs. $^1H$-NMR spectrum for $H_2O$ adsorbed on

GO with $w = 57$ mg g$^{-1}$ at $P/P_0 = 0.1$ (Fig. S9) shows that the $H_2O$ molecules with the lower chemical shift (i.e., 6.92 ppm) are not observed, indicating that strongly bound $H_2O$ molecules are predominant at the initial stage of adsorption.

With the measurement of $^1$H-NMR spectrum for the $D_2O$ (99.96 at% D, Cambridge Isotope Ltd.) adsorbed GO, we observed the hydrogen bonding information of HDO molecules surrounded by $D_2O$ molecules, indicating not direct information of hydrogen bond between $D_2O$ molecules in GO frameworks. However, $^2$H-NMR spectrum for $D_2O$ adsorbed on GO (Fig. S10) gives similar profile of the resonance line to that for $^1$H-NMR, suggests the applicability of the above discussion in Fig. 5b to the hydrogen bonding of $D_2O$.

Wide line $^2$H-NMR spectra for $D_2O$ adsorbed on GO at near the saturated adsorption ($w = 680$ mg g$^{-1}$) were measured using a spectrometer (Chemagnetics CMX Infinity) equipped by a superconducting magnet with an external magnetic field of $B_0 = 7.0$ T, which gives a Larmor frequency of 30.7 MHz for $^2$H nuclei. The free induction decay (FID) signals were recorded using the solid-echo pulse sequence, which comprises two π/2-pulses of 3-μs length, a pulse interval of 20 or 40 μs, and a repetition time of 4 s. The $^2$H-NMR spectra were measured at the temperature range from 153 K to 293 K within heating process. The sample temperature was controlled within the experimental error of ±1 K by the regulated $N_2$ gas flow.

## Modeling isotopic quantum effect in $D_2O$ versus $H_2O$
Quantum effects in water should be investigated by implementing the Feynman–Hibbs approach (QFH) using a reference potential for $H_2O$ and $D_2O$ published in the citation[41]. In this work, the evolution of the thermodynamics, the structure, the diffusivity, and the dynamics in light and heavy water is investigated over a large range of temperature and is compared with experimental data and with classical simulations as well. The accuracy of the results and the very low cost in computer time make the Feynman–Hibbs approach a valuable procedure to rapidly estimate the order of magnitude of the quantum contributions to intermolecular properties of water.

From the path-integral quantum partition function (without exchange) for a canonical ensemble ($N$, $V$, $T$) of atoms, and after some algebra, the FH potentials can be obtained. By keeping quadratic fluctuations around the classical path, one obtains the QFH potential ($U_{QFH}$):

$$U_{QFH}(r) = U_C(r) + \frac{\beta \hbar^2}{24 \mu} \left[ U_C''(r) + 2 \frac{U_C'(r)}{r} \right] \qquad (8)$$

This potential is built so as to improve upon a classical interaction model ($U_C$), normally Lennard-Jones (LJ) + Coulombic for $H_2O$ and $D_2O$, by taking into account factors related to quantum features (with $\hbar = h/2\pi$, $h$ the Plank constant, the effective mass $\mu = m_1.m_2/m_1 + m_2$, $\beta = 1/kT$, $k$ the Boltzmann constant, $T$ temperature). The estimate of the quantum effects by the QFH potential is only valid when the quantum corrections to classical quantities remain small. The order of magnitude of these corrections is given by the value of the parameter ($2\beta\hbar^2/m\sigma^2$) where $\sigma$ can be taken equal to the De Broglie wave length ($l = h/(2\pi mkT)^{1/2}$) that is a typical length associated to the size of system molecules or atoms, for instance, equal to the $\sigma$ parameter of the LJ potential modeling the interactions. For rare gas such as Ne, it is worthwhile to point out that at distances $r \leq r_m$ (LJ-minimum) the repulsive character of the QFH potential predicts a larger energy (a more positive repulsion at short distance) than the uncorrected potentials; this difference being more pronounced as $r$ and $T$ both decrease showing that the QFH potential tends to increase the apparent molecular size with some obvious implications regarding thermodynamics properties[42]. It is found that quantum effects are significant near ambient conditions and vanish

with increasing temperature less drastically than generally assumed. The most affected quantity is the self-diffusion coefficient.

In the context of water adsorption in carbon nanoporous materials, we used the interatomic potential forms to describe water–water interaction and the isotope deuterium effect following Guillot et al. reparametrizing the original Rahman-Stillinger central force water potential[43–45]. The interatomic forms for the water bonded (described by a Morse function) and non-bonded interactions (combining an electrostatic charge-charge $1/r$ form and dispersion interactions) are given by the following equations:

$$V_{OO}(r) = \frac{144.358}{r} + 0.5 \left[ \left( \frac{3.74}{r} \right)^8 - \left( \frac{3.74}{r} \right)^6 \right], \qquad (9a)$$

$$V_{OH}(r) = -\frac{72.269}{r} - \frac{4.3}{1 + e^{0.9(r-2.2)}} + 3.9 \left[ \left( e^{-5.8(r-1.07)} - 1 \right)^2 - 1 \right], \qquad (9b)$$

$$V_{HH}(r) = \frac{36.1345}{r} + \frac{17}{1 + e^{3.1(r-2.05)}} + 13 \left[ \left( e^{-6.0(r-1.495)} - 1 \right)^2 - 1 \right]. \qquad (9c)$$

We then calculated the quantum Hibbs Feynman correction regarding the H/D isotopic effect for the last two interatomic potential functions (Eqs. 9b and 9c) using Eq. 8 that requires evaluating the analytical first and second derivatives ($U_C'(r)$ and $U_C''(r)$) with respect to distance (evaluated using Chat-GTP and Wolfram-Alpha web applications) and the $\beta\hbar^2/24$ $m$ factor. At room temperature, their values for the H-H, D-D, H-O and D-O pairs are $1.94.10^{-2}$, $9.70.10^{-3}$, $1.02.10^{-2}$, $5.51$ $10^{-3}$ kcal mol$^{-1}$ respectively using the molar mass of O, H and D at 16, 1 and 2 g mol$^{-1}$. Figure S1 presents the change in the various interatomic potentials involving H(D) pairs at 300 K.

## Modeling of GO interlayer structure with MD simulations
We used the LAMMPS[46] software for molecular dynamics (MD) simulations with the ReaxFF[47] force field. The initial configuration of GO sheet structure was obtained from a $4.263 \times 3.938$ nm$^2$ graphene sheet by addition of functional groups at random. The atomic percentage of carbons for C-C/C = C, C-OH/C-O-C and C = O are 37, 44 and 19% in the GO sheet structure (Fig. S11), respectively, that we determined from X-ray photoelectron spectroscopy (XPS) measurements (JPS-9010TR JEOL co.) with MgKα (10 kV, 20 mA) source. The rGO model was constructed by removing oxygen functional groups from the model of the simply stacked ultra-large GO sheet structure. The atomic percentage of carbons for C-C/C = C, C-OH/C-O-C, and C = O are 89.1, 7.1, and 3.8% in the rGO sheet structure, respectively, that we determine from our previous XPS result for rGO obtained by heat treatment of GO at 873 K under Ar flow condition[48]. We used the random distribution of oxygen functional groups on GO layers in our simulation although the oxidation could take place near oxidized site, resulting in oxidized and unoxidized domain formations on GO[49]. There is some influence of the ordering of the models on the properties of the water near the interface, mostly for diffusion; but the differences are small and the overall behavior unchanged.

For the simply stacked GO model, four GO sheets were put in a simulation cell ($L_x = 4.263$ nm, $L_y = 3.938$ nm, and $L_z = 2.8$ nm) with an interlayer distance of 0.7 nm in the z-axis direction. Each GO sheet was alternately shifted by half the length of the simulation cell in the x- and y-axis directions and periodic boundary conditions were applied in all three directions. Since the lateral size of this GO model can be regarded as an infinite length due to the periodic boundary conditions, this calculation with the model is directly related to the important application of ultra-large GO sheets whose lateral size is over 10 μm with fewer edges and lower inter sheets junction structure. Therefore, we call this GO model as "ultra-large GO" in this study.

For the partially staggered stacking GO model, we made the twenty GO sheets (ca. 4.5 nm × 3.938 nm), which are finite in the $x$-direction by attaching functional groups on the edges, stacked in the $z$-direction with the distance of 0.7 nm. Each GO sheet was randomly shifted in the $x$- and $y$-axis directions. The size of the simulation box was $L_x = 8.026$ nm, $L_y = 3.938$ nm, and $L_z = 14.0$ nm, and the periodic boundary conditions were applied in all three directions. Using the two GO and the one rGO models, we performed MD simulation in the isothermal-isobaric ($NPT$) ensemble at 298 K and 0.001 atm. The $NPT$-MD runs were 100 ps with a time step of 0.125 fs.

We also constructed a simply stacked ultra-large GO model accommodating $H_2O$ molecules, which is the same as the simply stacked ultra-large GO model mentioned above except that the interlayer distance in the $z$-direction was set to 0.75 nm. The number of $H_2O$ molecules was determined from the $H_2O$ adsorption isotherm at $P/P_0 = 0.8$, and 888 molecules were randomly inserted between the GO sheets. The partially staggered stacking structure of GO with $H_2O$ molecules was constructed similarly to the previously described model, differing only in the layer spacing of 0.75 nm and the random insertion of 2220 molecules. Using the two GO models including $H_2O$ molecules, we performed MD simulation in the canonical ($NVT$) ensemble at 298 K with all the GO coordinates fixed. The $NVT$-MD runs were 10 ps with a time step of 0.125 fs and the ReaxFF force field was also applied for the $H_2O$ molecules. Following those, $NPT$-MD runs were conducted at 298 K and 0.025 atm for 100 ps with a time step of 0.125 fs.

We also performed additional MD simulations for the staggered stacking structure of GO with $H_2O$ molecules to reduce the length of simulation box $L_x$ form 8.026 nm to ca. 6.4 nm with the pressure in the $y$- and $z$- directions controlled to 0.025 atm. The obtained structure was equilibrated by the $NPT$-MD run of 50 ps at 298 K and 0.025 atm (Fig. 1b). After that, the $NPT$-MD run for the equilibrated structure of staggered stacking of GO with $H_2O$ molecules was performed for 10 ps at 298 K and 0.025 atm to determine the square displacement of $H_2O$ molecules. For the analysis of $H_2O$ dynamics on staggered stacking structure of GO, we randomly selected 85 of $H_2O$ molecules from our MD simulation box and the squared displacement of each $H_2O$ molecule at 10 ps was obtained from the plots of time versus squared displacement (Fig. 2b). The statistical distribution of the squared displacements at 10 ps is shown in Fig. 2c. Finally, the percentage of slower and faster $H_2O$ molecules was evaluated using the distribution curve.

The interlayer distances for simulated GO structures were assessed by following two ways: the one is dividing the cell length of $z$ axis by the layered number of GO sheets in the box after equilibrium and the other is the 001 peak position of calculated X-ray scattering intensity ($I(s)$) in Fig.1. $I(s)$ was evaluated from the simulated GO structures with the following equation:

$$I(s) = \sum_i^{i \neq j} \sum_j f_i f_j \frac{\sin s r_{ij}}{s r_{ij}}$$

where $r_{i,j}$ is the distance between two atoms $i$ and $j$, and $f_i$ is the atomic scattering factor of atom $i$. Both values of interlayer distances showed very good agreement each other, indicating the validity for the estimation of the interlayer distances from simulated GO structures.

## Reporting summary
Further information on research design is available in the Nature Portfolio Reporting Summary linked to this article.

## Data availability
The data that support the findings of this study are available from the corresponding author upon request. Source data are provided as Source Data file. Molecular dynamics data (the initial and final configurations) are available in Supplemental Data file. Source data are provided with this paper.

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

## Acknowledgements
R.F., H.T., and K.K. were supported by Kotobuki HD. Fund. This research was supported by Grant-in-Aid for Scientific Research (B) (No. JP22H01905), for Challenging Research (Exploratory) (No. JP21K19022), and JST Open Innovation Platform with Enterprise, Research Institute and Academia (OPERA)- (JPMJOP1722). This research was also partially supported by the New Energy and Industrial Technology Development Organization (P21005). The authors thank Dr. Naoya Inazumi and Dr. Yasuto Todokoro of the Analytical Instrument Facility, Graduate School of Science, Osaka University for their helpful and useful advice, guidance, and instructions concerning NMR measurements. We thank Ms. Nurul Chotimah, Ms. Austina Dwi, Mr. Akira Sakima, and Ms. Mami Wada for assisting in the synthesis of GO and TG-DTA experiments for GO adsorbing water. We also would like to thank Editage (www.editage. com) for English language editing.

## Author contributions
R.F. carried out the experiments, being in main charge of this research. R.F. and K.K. prepared the manuscript. T.U. conducted ex-situ NMR measurements. A.F. measured TEM images of GO. H.T. and P.A.B. performed MD simulations. T.I. contributed to develop in situ X-ray diffraction measurements. R.F., T.I., H.T., T.U., F.X.C., R.J.M.P., and K.K. discussed the results and edited the paper.

## Competing interests
The authors declare no competing interests.
