## [Peer Review File · Nature Communications]

Staggered structural dynamic-mediated selective adsorption of H₂O/D₂O on flexible graphene oxide nanosheetsREVIEWER COMMENTS

Reviewer #1 (Remarks to the Author):

The paper is intended to reveal microscopic mechanism about H₂O and D₂O separation, and it provides the understanding of water adsorption behaviour in the flexible GO framework, and microscopic insight on higher affinity for H₂O than D₂O near interfaces. The data related to MD simulations, in-situ XRD and ex-situ NMR are well presented. However, there are the following major issues: (1) The title does not match well with the content, as the paper focused on the adsorption behaviour of hydrogen isotopic water, rather than the separation of H₂O/D₂O. The adsorption is just one of the steps involved in the separation of hydrogen isotopic water, the work on other steps should be added if the current title is used. Moreover, the temperature range studied in the paper was not comparable with the referred literature on the separation of H₂O/D₂O, such as the reference 10. (2) Regarding the adsorption behaviour, the paper was based on the GO nanosheets prepared by the authors, but no detail about the physical information about GO nanosheets was provided, such as the lateral size and porosity of GO nanosheet. The current staggered structure may not exist or another type of structure may arise if the large or ultralarge lateral size of GO sheets were used, as it is known in the literature that GO sheet can be porous prepared by Hummers method, and hydrophilic functional groups exist because of these pores. Therefore, the detail or characterization data about GO nanosheets should be added.

If the title is changed to the one about the adsorption behaviour, and the more MD simulations based on (i) large or ultra-large GO nanosheets, (2) reduced GO (rGO) nanosheets are added, the paper can be more valuable for the research field about GO membranes.

There are other minor issues to be addressed, including (1) adding the sample preparation detail about TEM, (2) the detail about the way to obtain 58% in Figure 1.

Reviewer #2 (Remarks to the Author):

The focus of this paper is to determine the mechanism of the higher propensity of H₂O versus D₂O for the graphene oxide surface in order to aid the development of GO based separation techniques. The authors used a combination of experimental (XRD and NMR) and computational methods to carry out these investigations. The work is potentially very intriguing. However, there are a number of issues with the MD simulations carried out in this work.

1. The placement of the oxygenated groups on the GO sheet was done at random. Work by Sinclair et al. discusses this and in fact they have used experimental and theoretical calculations (that show that oxidation takes place near oxidized sites and in fact GO has oxidized and unoxidized domains).to come up with a model for the placement of the oxygenated groups (. Chem. Inf. Model. 2019, 59, 6, 2741–2745).
2. Work has shown the GO sheet is reactive including in the presence of water. In fact a co-author on this paper was a co-author on another paper that showed reactive from ab initio MD simulation.

(Nature Communications volume 11, Article number: 1566 (2020) .Furthermore, Rolf et al also saw reactivity in their ab initio MD simulations. (Chem. Commun., 2021,57, 11697-11700; J. Phys. Chem. B 2020, 124, 37, 8167–8178). However in this paper they use a force-field which is reactive (REAXFF) but they do not discuss the reactivity. Furthermore, they do not discuss any kind of validation of the force-field for the graphene oxide system.. Isotope effects on the reactivity is well known in other systems. It could be important here as well

3. If one truly wanted to evaluate nuclear quantum effects, one should use path integral simulations like Centroid MD. It is not clear that an effective force-field for water will incorporate these important nuclear quantum effects.

Reviewer #3 (Remarks to the Author):

In this paper, the authors combine atomistic simulations and microscopy/characterization techniques to investigate the behavior of GO in interaction with water isotopes. The authors aim to address fundamental concepts such as the difference between experimentally measured GO interlayer spacing and atomistic simulations, and the preferential adsorption of water isotopes at the functional groups of GO.

The subject is very interesting, and the paper is very well written and structured. The results are clearly explained and in the main manuscript descriptive figures accompany the claims.

Despite these positive aspects of the paper, there are several major points that must first be addressed, without which the conclusions and claims made in the paper are not sufficiently supported.

Major points:

1- On page 3 of the paper the authors claim:

adsorption of D₂O having a stronger hydrogen bonding between D₂O molecules occurs at only the optimum arrangement of surface functional groups, whereas H₂O of the weaker hydrogen bonding nature can adsorb at any surface functional group sites

This implies that in the competition of forming bonds between D₂O and D₂O-GO, stronger D₂O bonds result in less adsorption of D₂O at the functional groups. The question is why at very low pressures, where the interaction between the water molecules (H₂O and D₂O) is almost negligible, the adsorption of H₂O at the functional groups is still dominant?

2- The authors specified ref. 33 for ReaxFF potential, but this potential does not include Deuterium. This needs clarification.

3- The authors use quantum simulations to compare the behavior of water isotopes in interaction with GO. However, there are interatomic potentials such as the one below that could be used for this purpose. The readership would benefit to have this point addressed, why an interatomic potential such as the following was not used?

Isotope Effects in Water: Differences of Structure, Dynamics, Spectrum, and Proton Transport between Heavy and Light Water from ReaxFF Reactive Force Field Simulations Zhang, Weiwei; Chen, Xing; van Duin, Adri C. T.

4- One practical aspect of heavy water filtration is the unavoidable formation of HDO due to proton exchange. It will be very helpful if the behavior of DHO in interaction with GO discussed.

5- One major point that needs clarification is the contradiction of the conclusion of this work with the dominant hypothesis in the literature. In this work the authors report a weaker bond of D2O-GO compare to H2O-GO. In the literature (for instance references 10-12 of the paper), selective separation of heavy water during filtration is postulated to be due to stronger bond between D2O and functional groups and slower permeability of D2O compare to H2O. If the results of this work are not in agreement with the previous MD simulations and theories, what is their alternative explanation on the selective separation of heavy water?

Considering these points, I believe it is essential to address the raised points before I can recommend this work for publication.

We are deeply grateful to all the reviewers for the insightful comments and suggestions and for giving of their precious time to help us improve our manuscript. We have carefully considered their comments and suggestions in the revision of our manuscript.

Please find below our responses (in blue font) to each of the reviewer comments (in black font). Excerpts of the information added to our manuscript are presented in red font.

Reviewer #1

Comment (1)

The title does not match well with the content, as the paper focused on the adsorption behaviour of hydrogen isotopic water, rather than the separation of H₂O/D₂O. The adsorption is just one of the steps involved in the separation of hydrogen isotopic water, the work on other steps should be added if the current title is used. Moreover, the temperature range studied in the paper was not comparable with the referred literature on the separation of H₂O/D₂O, such as the reference 10.

Answer for comment (1) from authors

We appreciate the Reviewer's valuable comment and agree with their observation.

We have revised the title to "Staggered structural dynamics mediated water isotope **selective adsorption** on flexible graphene oxide nanosheets" to explicitly reflect the objective of the study. We carried out an additional experiment on the mixed gas adsorption using the mixed vapor of D₂O and H₂O using a newly constructed laboratory-designed equipment with a mass spectrometer to show the H₂O selective adsorption behavior of GO (Fig. S2 (a)).

However, the complex comparison between single component adsorption and mixed vapor adsorption must be avoided owing to the predominantly formed HDO in the H₂O-D₂O mixed phase because the chemical exchange occurs based on the following equilibrium constant:

$$K = \frac{[\text{HDO}]^2}{[\text{H}_2\text{O}][\text{D}_2\text{O}]}$$
$$= 3.85 \text{ (at 298 K)}$$

Therefore, we determined the H/D ratio of water isotopic mixtures through mass intensity measurements and compared it with the H/D value obtained from single component adsorption measurements. The vapor of the bulk mixture of H₂O: D₂O with a mole ratio 1.13 :1.00 (*i.e.*, H/D = 1.13) was adsorbed by GO at 298 K and $P/P_0 = 0.87$ for 24 h. We determined the composition of the mixed adsorbed layer on GO in the liquid state as

follows: we desorbed the adsorbed mixed layer on GO at 333 K for 2 h, collected the desorbed water isotopic mixture using a cold trap at 77 K, and then converted it to liquid at 298 K, for which the H/D ratio was determined using mass spectroscopy. Fig. 2S(b) shows the time courses of the $m/z = 18$ to $m/z = 20$ intensity-ratios (I^{18}/I^{20}) for the feed vapor (red) and desorbed vapor after adsorption on GO. The I^{18}/I^{20} of the adsorbed mixture clearly exceeds that of the feed vapor, indicating relatively high H contents in the adsorbed layer on GO. We determined the H/D ratios from the averaged I^{18}/I^{20} values for the last 100 s of these curves. The H/D ratios are 1.13 for the initial mixed vapor and 1.70 for the adsorbed layer, indicating selective adsorption of H₂O on GO.

The corresponding adsorption amounts of H₂O, D₂O, and HDO at $P/P_0 = 0.87$ were 8.36, 2.91 and 9.68 mmol/g, respectively, with $K = 3.85$ at 298 K. The adsorption amounts of H₂O and D₂O at $P/P_0 = 0.87$ for the adsorption branch of the single component adsorption isotherm were 11.5 and 9.48 mmol/g, respectively. This corresponds to H/D = 1.21 being a smaller isotopic difference than that observed for the mixed vapor adsorption (Table S1). This increased H/D ratio in mixed vapor adsorption indicates that the D₂O-phobisity of GO is emphasized by the existence of H₂O compared to that from single component adsorption because of the competitive adsorption on the unfavorable adsorption site of oxygen functional groups for D₂O.

To clarify this claim, we have added the following information on Page 4, line 9:

“Moreover, to confirm the isotopic selective adsorption characteristics of GO, the mixed vapor adsorption of D₂O and H₂O was conducted using laboratory-designed equipment with a mass spectrometer (see Fig S2(a)).

However, the complex comparison between single component adsorption and mixed vapor adsorption was avoided owing to the predominantly formed HDO in the H₂O-D₂O mixed gas phase as the chemical exchange occurs based on the following equilibrium constant:

$$K = \frac{[\text{HDO}]^2}{[\text{H}_2\text{O}][\text{D}_2\text{O}]} \\ = 3.85 \text{ (at 298 K)}$$

Therefore, we determined the H/D ratio of water isotopic mixtures using mass intensity measurements and compared it with the H/D value obtained from single component adsorption measurements. The vapor of the bulk mixture of H₂O: D₂O with a mole ratio of 1.13 :1.00 (*i.e.*, H/D = 1.13) was adsorbed by GO at 298 K and $P/P_0 = 0.87$ for 24 h. We determined the composition of the mixed adsorbed layer on GO in the liquid state as follows: we desorbed the adsorbed mixed layer on GO at 333 K for 2 h, collected the

desorbed water isotopic mixture using a cold trap at 77 K, then converted it to liquid at 298 K, for which the H/D ratio was determined by mass spectroscopy. Fig. 2S(b) shows the time courses of the $m/z = 18$ to $m/z = 20$ intensity-ratios (I^{18}/I^{20}) for the feed vapor (red) and the desorbed vapor after adsorption on GO. The I^{18}/I^{20} of the adsorbed mixture is clearly larger than that of feed vapor, indicating the relatively high H contents in the adsorbed layer on GO. We determined the H/D ratios from the averaged I^{18}/I^{20} values for the final 100 s of these curves. The H/D ratios are 1.13 for the initial mixed vapor and 1.70 for the adsorbed layer, respectively, indicating selective adsorption of H₂O by GO.

The corresponding adsorption amounts of H₂O, D₂O, and HDO at $P/P_0 = 0.87$ were 8.36, 2.91, and 9.68 mmol/g with $K = 3.85$ at 298 K, respectively. The adsorption amounts of H₂O and D₂O at $P/P_0 = 0.87$ for the adsorption branch of the single component adsorption isotherm are 11.5 and 9.48 mmol/g, respectively. This corresponds to H/D = 1.21 being a smaller isotopic difference than that observed for the mixed vapor adsorption (Table S1). This increment in the H/D ratio in mixed vapor adsorption indicates that the D₂O-phobisity of GO is emphasized by the existence of H₂O compared to that from single component adsorption because of the competitive adsorption on the unfavorable adsorption sites of oxygen functional groups for D₂O.”

Fig.S2 H₂O selective adsorption experiment of isotopic mixed water vapors on GO |

(a) Experimental setup for the isotopic mixture adsorption on GO using mass spectroscopy. (b) The time courses of the mass spectra for the feed vapor (red) and the desorbed vapor after adsorption on GO at $P/P_0 = 0.87$ for 1days (blue). The intensity ratios for $m/z = 18$ and 20 are shown.

Table S1 Adsorption amounts of water isotopes on GO in the single-component and mixed vapor adsorption

	Mixed vapor adsorption	Single component adsorption
H ₂ O (mmol/g)	8.36	11.5
D ₂ O (mmol/g)	2.91	9.48
HDO (mmol/g)	9.68	-
H/D	1.70	1.21

Comment (2)

Regarding the adsorption behaviour, the paper was based on the GO nanosheets prepared by the authors, but no detail about the physical information about GO nanosheets was provided, such as the lateral size and porosity of GO nanosheet. The current staggered structure may not exist or another type of structure may arise if the large or ultralarge lateral size of GO sheets were used, as it is known in the literature that GO sheet can be porous prepared by Hummers method, and hydrophilic functional groups exist because of these pores. Therefore, the detail or characterization data about GO nanosheets should be added.

Answer for comment (2) from authors

We measured TEM images for additional samples and carefully analyzed these data to obtain reliable information on the staggered structure. Furthermore, we measured the Raman spectrum to determine the carbon structure. We have added newly obtained characterized data on the GO layer to the manuscript to expound on the characterization of GO based on TEM, X-ray diffraction (Fig. S3(a) and Table S2), and Raman spectroscopic data (Fig. S3(b) and Table S3).

Figure 2(a) shows the TEM image of stacking GO layers, highlights the lateral size with yellow lines, and the lateral size distribution curve (Fig. 2(c)) indicates that the average GO size is approximately 2 nm. Evidently, wrinkling GO layers make up the stacked structures, resulting in staggered structures having relatively widened interlayer spaces, as shown by the red arrows in Fig. 2 (b). Our statistical analysis for the TEM images gives 0.8 nm^{-2} of the density of staggered and widened interlayer structure as shown in Fig.2 (d) with red dots.

The lateral size of graphene-like sheet structures was also characterized by the G and D band intensity ratio of the Raman spectrum (*i.e.*, $L_a = 12 \text{ nm}$) and by the width of 10 and 11 diffraction peaks (*i.e.*, the crystallite sizes of $L_{10} = 11 \text{ nm}$ and $L_{11} = 6.5 \text{ nm}$) of the

XRD result as shown in Tables S2 and S3. These results indicate that the lateral size of graphene-like sheets is ca. 10 nm.

The various structure characterizations indicate that the 2-nm GO layer is a primitive unit as observed in the stacking structure determined using TEM measurements, thereby underestimating the lateral size of graphene sheets owing to the limited focus regions for TEM observation. In fact, the primitive units of GO layers may be sequentially combined to form a larger unit as shown by XRD and Raman spectroscopy. In our MD simulation of the staggered layered GO, the length of GO sheets in the x -axis is c.a. 4.5 nm, which is comparable to the primitive unit size of GO sheets. The primitive unit size of GO layers should play a significantly important role in the structural flexibility of the staggered structures.

Therefore, we have added the analyzed TEM images (Figure 2) and the following information on Page 3, line 11.

Furthermore, we described the staggered structure of realistic GO through the precise analysis of high-resolution TEM images. Figure 2(a) shows the TEM image of stacking GO layers, highlights the lateral size with yellow lines, and the lateral size distribution curve (Fig. 2(c)) indicates that the average GO size is approximately 2 nm. Evidently, wrinkling GO layers form the stacked structures, resulting in staggered structures with relatively widened interlayer spacing, as shown by the red arrows in Fig. 2 (b). Based on statistical analysis of the TEM images, the density of the staggered and widened interlayer structure is 0.8 nm^{-2} as shown by the red dots in Fig. 2 (d).

The lateral size of the graphene-like sheet structures was also characterized using the G and D band intensity ratio of the Raman spectrum (*i.e.*, $L_a = 12 \text{ nm}$) and by the width of 10 and 11 diffraction peaks (*i.e.*, the crystallite sizes of $L_{10} = 11 \text{ nm}$ and $L_{11} = 6.5 \text{ nm}$) of the XRD result as shown in Tables S2 and S3. These results indicate that the lateral size of graphene-like sheets is ca. 10 nm.

The various structure characterizations indicate that the 2-nm GO layer is a primitive unit as observed in the stacking structure determined using TEM measurements, thereby underestimating the lateral size of graphene sheets owing to the limited focus regions for TEM observation. In fact, XRD and Raman spectroscopy showed that the primitive units of GO layers may be sequentially combined to form relatively large units. In our MD simulation of the staggered GO layers, the length of GO sheets in the x -axis was c.a. 4.5 nm, which is comparable to the primitive unit of GO sheets. The primitive unit size of GO layers is an essential influencing factor of the structural flexibility of the staggered structures.

Figure. 2 High-resolution TEM images of the staggered structures in GO| (a) the GO layer structure highlighted the lateral size with yellow lines and (c) the lateral size distribution. (b) The staggered and relatively widened interlayer spaces as shown by red arrows. (d) The staggered structures in the layers with the density of 0.8 nm^{-2} , as shown by red dots.

Fig. S3 (a)XRD and (b)Raman spectrum of GO

Table S2 The *d*-spacing and crystalline size of GO obtained from XRD

	2 θ (degree)	d (nm)	L (nm)
(001)	5.65	0.72	4.7
(10)	19.1	0.21	11.0
(11)	33.3	0.12	6.5

Table S3 D and G-bands of Raman spectrum for GO

D band (cm ⁻¹)	G band (cm ⁻¹)	I _D / I _G	L _a (nm)
1360	1613	1.72	12

Comment (3)

If the title is changed to the one about the adsorption behaviour, and the more MD simulations based on (i) large or ultra-large GO nanosheets, (2) reduced GO (rGO) nanosheets are added, the paper can be more valuable for the research field about GO membranes.

Answer for comment (3) from authors

As suggested by the reviewer, we performed additional MD simulations with ultra-large GO and rGO model developed via the removal of oxygen functional groups from the GO model.

The experimentally determined-carbon atomic percentages of the rGO model for C-C/C=C, C-OH/C-O-C, and C=O were 89.1, 7.1 and 3.8% in the rGO sheet structure, respectively. We determined the surface oxygen groups using XPS for rGO prepared by heating GO at 873 K under Ar (S. Wang et al. *Carbon* **2014**, *76*, 220-231). For both

simulations, several layers of the model sheets were placed in a simulation cell ($L_x = 4.263$ nm, $L_y = 3.938$ nm) with an interlayer distance of 0.7 nm in the z -axis direction. Each model sheet was alternately shifted by half the length of the simulation cell in the x - and y -axis directions and periodic boundary conditions were applied in all three directions. Using these models, we performed MD simulation in the isothermal-isobaric (NPT) ensemble at 298 K and 0.001 atm. The NPT -MD runs were 100 ps with a time step of 0.125 fs.

The simulated sheet structures of the ultra-large GO and rGO layers after equilibrium on MD simulation are shown in Fig. S4(a) and the layered structures in Fig. S4(b). The average interlayer spacings of the ultra-large GO and rGO were 0.60 and 0.33 nm, respectively. The interlayer spacing of rGO is quite close to that of graphite. Water molecules can adsorb in ultra-large GO interlayer spaces by swelling the interlayer distance to 0.75 nm.

Conversely, the rGO has a robust layer to layer van der Waals attractive interaction, making it difficult for H_2O molecules to exfoliate the interlayer structure, particularly for hydrophobic rGO, leading to a minor adsorption of H_2O . The observed swelling of GO on H_2O adsorption should originate from the oxygen-rich sheet structure differing from rGO.

We added the following information to the manuscript to explain these additional MD simulation results (Page 2, line 21):

“Figure 1(a) shows models of simply stacked ultra-large GO sheets with and without adsorbed water from MD simulations. Although the GO interlayer spacing increases with increasing water adsorption owing to the typical swelling nature of GO, the interlayer spacings are 0.1 and 0.3 nm smaller than the experimental ones identified via X-ray diffraction (XRD) with and without water adsorption, respectively (Figs.1(c) and 3). We also studied the effects of functional groups via the MD simulation of the reduced graphene oxide (rGO) model, developed by removing oxygen functional groups from the ultra-large GO structure (Methods). The simulated sheet structures of the GO and rGO layers on MD simulations after equilibrium are shown in Fig. S4(a) and the layered structure in Fig. S4(b). The average interlayer spacings of the GO and rGO were 0.60 and 0.33 nm, respectively. The interlayer spacing of rGO is similar to that of graphite. Water molecules can be adsorbed in the GO interlayer spaces causing the interlayer to swell to 0.75 nm.

Conversely, the rGO exhibit a robust layer to layer van der Waals interaction, making it difficult for H_2O molecules to exfoliate the interlayer structure, particularly

for hydrophobic rGO, leading to a minor adsorption of H₂O. The observed swelling of GO following H₂O adsorption is attributed to the difference between the oxygen-rich sheet structure and rGO.”

”

Fig. S4 (a) Snapshots of the sheet structure of a ultra-large GO layer and rGO layer after equilibrium on MD simulation. **(b)** Snapshots of the layered structures of each model after equilibrium on MD simulation. Here, we constructed the ultra-large GO and rGO layer models using the XPS data. We used the carbon atomic percentages for C-C/C=C, C-OH/C-O-C and C=O are 37, 44 and 19% in the GO sheet structure. The rGO model was constructed by removing oxygen functional groups from the model of simple stack GO layer structure. The atomic percentage of carbons for C-C/C=C, C-OH/C-O-C and C=O are 89.1, 7.1 and 3.8% in the rGO sheet structure.

Comment (4)

There are other minor issues to be addressed, including (1) adding the sample preparation detail about TEM, (2) the detail about the way to obtain 58% in Figure 1.

Answer for comment (4) from authors

We added the following sentences on page 9, line 19 to clarify point (1) and on page 12, line 25 to clarify (2).

(1) “The dried GO powder was lightly pressed on a 150 mesh Cu microgrid with carbons (Okenshouji Co., Ltd.) for the TEM observation; the solvent free method is preferable to avoid the morphological change of GO particles upon wetting and/or dispersion.”

(2) “For the analysis of water dynamics on GO, we randomly selected 85 water molecules from our MD simulation box and the squared displacement of each water molecule at 10 ps was obtained from the plots of time vs squared displacement (Fig. 1(e)). The statistical distribution of the squared displacements at 10 ps is shown in Figure 1(f). Finally, the percentage of slower water molecules was evaluated using the distribution curve.”

Reviewer #2

Comments (1)

The placement of the oxygenated groups on the GO sheet was done at random. Work by Sinclair et al. discusses this and in fact they have used experimental and theoretical calculations (that show that oxidation takes place near oxidized sites and in fact GO has oxidized and unoxidized domains).to come up with a model for the placement of the oxygenated groups (. Chem. Inf. Model. 2019, 59, 6, 2741–2745).

Answer for comment (1) from authors

We previously studied the influence of two types of models: random and semi-ordered (with domains) (F. Mouhat, F.-X. Coudert, M.-L. Bocquet, *Nature Commun.* **2020**, *11*, 1566). We found no strong, direct experimental evidence of the exact nature of domains in GO, which may depend heavily on the method of production. We found some influence of the ordering of the models on the properties of the water near the interface, mostly for diffusion; but the differences are small, and the overall behavior remains unchanged. To explain this point, we added the following sentences on Page 11, line 12:

“Here, we used the random distribution of oxygen functional groups on GO layers in our simulation although the oxidation could take place near oxidized site, resulting in oxidized and unoxidized domain formations on GO⁴⁰. There is some influence of the ordering of the models on the properties of the water near the interface, mostly for

diffusion; but the differences are small and the overall behavior unchanged.”

Comment(2)

Work has shown the GO sheet is reactive including in the presence of water. In fact a co-author on this paper was a co-author on another paper that showed reactive from ab initio MD simulation. (Nature Communications volume 11, Article number: 1566 (2020) .Furthermore, Rolf et al also saw reactivity in their ab initio MD simulations. (Chem. Commun., 2021,57, 11697-11700; J. Phys. Chem. B 2020, 124, 37, 8167–8178). However in this paper they use a force-field which is reactive (REAXFF) but they do not discuss the reactivity. Furthermore, they do not discuss any kind of validation of the force-field for the graphene oxide system. Isotope effects on the reactivity is well known in other systems. It could be important here as well

Answer for comment (2) from authors

As the reviewer commented, one of the co-authors conducted the ab initio MD simulation (F. Mouhat, F.-X. Coudert, M.-L. Bocquet, *Nature Commun.* **2020**, *11*, 1566). In their study, the authors showed that the reaction was possible, but it was not always observed (only one event was identified out of the trajectories for 6 different models). It was in most cases a local event, displaying a few proton transfers and producing transient charged species. The vast majority of the {GO + water} system was found to be unreactive.

In our MD simulation for GO, we selected the Reactive FF reported by Chenoweth, the most widely used ReaxFF for the hydrocarbon reaction simulations (K. Chenoweth & W. van Duin *J. Phys. Chem. A* **2008**,*112*, 1040-1053). Here, we also observed certain decomposition of hydroxyl groups and the hydrogen transfer from a physical adsorbed water to a hydroxyl group but they are not major events as shown in Fig. S5. In the snapshot, the reacted products are shown by CPK model without physically adsorbed water molecules. Only 7 OH⁻ ions or H₂O molecules were formed by the reactions, out of 888 molecules of physisorbed water molecules, after 100 ps of MD simulation.

Fig. S5 Snapshot of reactions for functional groups and water molecules in our MD simulation after 100ps calculation. Here, the products of water or hydroxyl groups are only shown as CPK models and the physisorbed water molecules are omitted for simplicity.

Although we lack knowledge on isotope effects on the reactivity because we did not perform a D₂O simulation, Sofer *et. al.* reported the exchange of deuterium with hydrogen in GO when GO materials are soaked in liquid D₂O under ultrasonication for 1 day. They showed that the increase in D contents in the GO was very limited (0.3 atomic %) by their treatment and exchange between D and H hardly occurs only on carboxylic acid group by highly lipophilic character of the hydroxyl groups where no hydrogen/deuterium exchange takes place (Z. Sofer *et. al.*, *ACS Nano* **2015**, *9*, 5478–5485). Based on this information, we imagine the transfer of deuterium, such as ¹H proton transfer, to be a rare event. To explain this point, we have added the following sentences on Page 8, line 12.

“Moreover, the decompositions of the functional group are very important for molecular separation with GOs. The ab initio MD simulation by Mouhat *et al.* showed that the decomposition of GO functional groups was seldom observed and most of the system was unreactive.¹⁴ In our MD simulation, we observed certain decompositions of hydroxyl functional groups and the hydrogen transfer from a physical-adsorbed water to a hydroxyl functional group; however, these are not major events as shown in Fig. S5.”

Comment (3)

3. If one truly wanted to evaluate nuclear quantum effects, one should use path integral simulations like Centroid MD. It is not clear that an effective force-field for water will incorporate these important nuclear quantum effects.

Answer for comment (3) from authors

Actually, we did not calculate the quantum effects in our MD simulation. As the reviewer pointed out, the differences between the fluidic behaviors of H₂O and D₂O should be considered from the quantum MD simulation. Based on our experience with respect to path integral quantum MD simulations of H₂ and D₂ molecules within very simple pore geometry (Quantum Effects on Hydrogen Isotopes Adsorption in Nanopores, H. Tanaka, D. Noguchi, A. Yuzawa, T. Kodaira, K. Kaneko, *J. Low Temp. Phys.* (Invited to a special issue), **157**, 352-373 (2009)), the use of this approach for the complicated geometry of the solid interface of GO models proposed in our study is impractical and time consuming. Although we believe the uniqueness of water in GO interlayer spaces should be associated with the quantum effects of water, we cannot address this important subject speedily. Therefore, we hope to perform this rigorous quantum simulation for H₂O and D₂O on GO in subsequent research. We believe this study should be published as soon as possible owing to its significance. To clarify this point, we have added the following information on Page 8, Line 26:

“Although we simulated the microscopic behaviors of H₂O molecules in the GO interlayer spaces based on MD with reactive FF, the observed differences between the fluidic behaviors of H₂O and D₂O can be elucidated more clearly from the quantum effects with the aid of path integral simulations like Centroid MD.^{31,41} A detailed study should be conducted with path integral quantum MD simulation with the realistic staggered GO structures as in our near future works.”

Reviewer #3

Comment (1)

On page 3 of the paper the authors claim:

adsorption of D₂O having a stronger hydrogen bonding between D₂O molecules occurs at only the optimum arrangement of surface functional groups, whereas H₂O of the weaker hydrogen bonding nature can adsorb at any surface functional group sites

This implies that in the competition of forming bonds between D₂O and D₂O-GO,

stronger D₂O bonds result in less adsorption of D₂O at the functional groups. The question is why at very low pressures, where the interaction between the water molecules (H₂O and D₂O) is almost negligible, the adsorption of H₂O at the functional groups is still dominant?

Answer for comment (1) from authors

As the reviewer mentioned, the stronger hydrogen bonding interaction of D₂O molecules results in reduced adsorption of D₂O in the GO interlayer spacing where water molecules are frustrated to form hydrogen bonding networks. Therefore, the difference between the adsorption amounts of H₂O and D₂O molecules of GO increased as adsorption increased (Figure S6(a)). However, the effects of frustrated hydrogen bonding networks in GO interlayer spacing are not limited to the highly adsorbed state but also to the initial stage of water adsorption. Figure S6(b) shows the magnified images of H₂O and D₂O adsorption isotherms of GO at very low $P/P_0 < 0.03$. The adsorption amounts of H₂O exceed that of D₂O at any relative pressure even in this very low-pressure region and the difference is equivalent to 12 % of the adsorption amounts at $P/P_0 = 0.024$, indicating the clear D₂O-phobicity of GO functional groups.

This is not only explained by the quantum effects of water isotopes but also by the surface structure of GO. Predominant water molecules are two-fold coordinated hydrogen atoms bonded with surface functional groups on GO. An H₂O molecule has more optimized adaptability to interact with the surface functional groups on GO through their excellent hopping motion. Furthermore, H₂O molecules can more easily form non-H-bonds between themselves than D₂O molecules, including in the GO interlayer spaces, because of the enhanced localization of heavy D atoms. As a result, the D₂O molecule is less adaptable to the strongly constrained local arrangement of surface functional groups on the GO, resulting in a smaller adsorbed amount for the observed D₂O-phobicity than H₂O even at low P/P_0 . To clarify this point, we added the following information on Page 7, Line 23 as follows:

“The difference between the adsorption amounts of H₂O and D₂O molecules of GO increases as adsorption progresses, but non-negligible preferential adsorption of H₂O is observed in the initial stage of water adsorption as shown in the magnified figures of H₂O and D₂O adsorption isotherms of GO at very low $P/P_0 < 0.03$ (Fig. S6(b)). The adsorption amount of H₂O exceeds that of D₂O at all pressure levels, including this very low-pressure region and the difference is equivalent to 12 % of the adsorption amounts at $P/P_0 = 0.024$, indicating the clear D₂O-phobicity of GO functional groups. This is not only explained

by the quantum effects of water isotopes but also by the surface structure of GO.

We propose a microscopic adsorption mechanism (Fig. 5(b)). Predominant water molecules are two-fold coordinated hydrogen atoms bonded with surface functional groups on GO. H₂O molecules exhibits more optimized interaction with the surface functional groups on GO through their excellent hopping motion. Furthermore, H₂O molecules can more easily form non-H-bonds between themselves than D₂O molecules, including in the GO interlayer spaces, owing to the enhanced localization of heavy D atoms³⁰. As a result, the D₂O molecule is less adaptable to the strongly constrained local arrangement of surface functional groups on the GO, giving the observed D₂O-phobicity a smaller adsorbed amount than H₂O even at low P/P_0 .”

Fig. S6 (a) Vapor adsorption isotherms of H₂O and D₂O on GO at 298 K. The vertical axis is expressed with the unit of mmol g^{-1} . (b) Magnified figures of the isotherms at low $P/P_0 < 0.03$.

Comment (2)

2- The authors specified ref. 33 for ReaxFF potential, but this potential does not include Deuterium. This needs clarification.

Answer for comment (2) from authors

In this study, we do not treat D₂O molecules with our reactive MD simulation. We identified the staggered interlayer GO structure to elucidate structural changes in GO upon water adsorption. We then studied H₂O adsorption by GO even with molecular dynamics. As the staggered interlayer GO structure can facilitate flexible and hydrophilic

nanoscale environments for hydrogen bonding formation with H₂O, we conjecture their selective adsorption property for H₂O and D₂O, as suggested by our Feynman–Hibbs (FH) effective potential approaches. Fortunately, we identified the remarkable H₂O adsorption selectivity of the staggered structured GO.

As the reviewer pointed out, the differences between the fluidic behaviors of H₂O and D₂O should be considered in MD simulation with reactive force fields including deuterium or from the quantum MD simulation. We have in the past performed path integral quantum MD simulations of H₂ and D₂ molecules within very simple pore geometry (Quantum Effects on Hydrogen Isotopes Adsorption in Nanopores, H. Tanaka, D. Noguchi, A. Yuzawa, T. Kodaira, K. Kaneko, J. Low Temp. Phys.(Invited to a special issue), 157, 352-373 (2009)). However, this approach is time consuming and impractical for the complicated geometry of the solid interface of GO models proposed in our study. Although we believe the uniqueness of water in GO interlayer spaces should be associated with the quantum effects of water, we cannot address this important subject speedily. Therefore, we hope to perform this rigorous quantum simulation for H₂O and D₂O on GO in our subsequent research. We believe this study should be published as soon as possible owing to its the importance. To clarify this significance, we have added the following information on Page 8, Line 18:

“In this study, we do not treat D₂O molecules with our reactive MD simulation because our FF does not include deuterium.”

Comment (3)

3- The authors use quantum simulations to compare the behavior of water isotopes in interaction with GO. However, there are interatomic potentials such as the one below that could be used for this purpose. The readership would benefit to have this point addressed, why an interatomic potential such as the following was not used?

Isotope Effects in Water: Differences of Structure, Dynamics, Spectrum, and Proton Transport between Heavy and Light Water from ReaxFF Reactive Force Field Simulations Zhang, Weiwei; Chen, Xing; van Duin, Adri C. T.

Answer for comment (3) from authors

In this paper, we do not regard D₂O molecules in MD simulation as earlier mentioned. We identified the staggered interlayer GO structure to elucidate structural changes in GO upon water adsorption. We then studied H₂O adsorption on GO through molecular

dynamics. Herein, we selected the Reactive FF reported by Chenoweth, which is the most widely used ReaxFF for reactive hydrocarbon simulations (K. Chenoweth & W. van Duin *J. Phys. Chem. A* **2008**,112, 1040-1053). The study has been cited more than 1700 times, indicating the high reliability and state-of-the-art level of this force field.

As the reviewer recommended, the mechanism underlying the isotopic adsorption difference between H₂O and D₂O in the staggered layer structure can be elucidated through MD simulations using the force field proposed by Zhang *et al.* Although this recently proposed force field would be suitable in our study, we can only use it to elucidate the isotopic adsorption difference in GO after performing systematic studies on the availability of the proposed force field for adsorption behaviors in various carbon nanopores. In addition, restarting our MD simulation of the water adsorption on the staggered structure of GO with this force field may take more time: we regard this as scope for future research. Therefore, we only mention the usefulness of the force field to elucidate isotopic differences between H₂O and D₂O as mentioned on Page 8, Line 19:

“Recently, Zhang *et al.* proposed a reactive force field that can adequately characterize the differences in the radial distribution function (RDF), self-diffusion constant, and vibrational spectrum between heavy and light water and is suitable for elucidating the water isotopic differences in the GO structure with MD simulation.^{32, 11”}

Comment (4)

4- One practical aspect of heavy water filtration is the unavoidable formation of HDO due to proton exchange. It will be very helpful if the behavior of DHO in interaction with GO discussed.

Answer for comment (4) from authors

While this interaction is important, the complexity of existing HDO molecules in the isotopic mixture makes the quantification of their separation difficult. Upon mixing the D₂O and H₂O, the chemical exchange occurs with the equilibrium constant $K = \frac{[\text{HDO}]^2}{[\text{H}_2\text{O}][\text{D}_2\text{O}]} = 3.85$ at 298 K in bulk. For the mixtures in separation membranes or in solid adsorbents, this value can be changed because the interactions between water isotopes and the substances would differ from each other from the viewpoints of the hydrogen bond. The MD simulation of the mixture of H₂O and D₂O in the interlayer spaces of GO is expected to provide valuable insights for the formation of HDO from microscopic viewpoints. However, we focused on the anomalous behavior of water in the GO interlayer spacing and the aforementioned experiments would exceed the scope of our

study. Therefore, we have added the following information on Page 8, line 8:

“For the practical use of GO as water isotopic separation filters, the presence of HDO is inevitable owing to proton exchange. Furthermore, the equilibrium constant K could differ from that of bulk value because the interactions between the water isotopes and substances differ.”

Comment (5)

5- One major point that needs clarification is the contradiction of the conclusion of this work with the dominant hypothesis in the literature. In this work the authors report a weaker bond of D₂O-GO compare to H₂O-GO. In the literature (for instance references 10-12 of the paper), selective separation of heavy water during filtration is postulated to be due to stronger bond between D₂O and functional groups and slower permeability of D₂O compare to H₂O. If the results of this work are not in agreement with the previous MD simulations and theories, what is their alternative explanation on the selective separation of heavy water?

Answer for comment (5) from authors

As mentioned by Reviewer 1, the inconsistency between the conclusion of our study and that of the references is attributed to the difference in the mechanism underlying the adsorption separation process (in this study) and membrane separation process (References 10 and 11). The pervaporation separation with membrane filters is related to numerous factors such as gas permeation or adsorption to the solid, gas diffusion by the difference in gas pressure or concentration, and gas desorption from the solid. Conversely, the adsorption separation process constitutes one of the steps and the most important factor is the interaction between the gas molecules and solids. Furthermore, pervaporation is conducted at high temperature (c.a. 373 K in Reference 10) to permeate gas molecules to solid faster even when our adsorption separation of water isotopes was conducted at 298 K. Generally, the quantum effects of water isotopes are significant at ambient temperature and vanish with increasing temperature. At this point, the D₂O selective filtration by pervaporation may be primarily caused by the kinetic reason underlying the heavier isotope of D₂O and not quantum effects.

To clarify this point, we have added the following information on Page 5, Line 6:

“These results are inconsistent with those of the pervaporation membrane separation reported by Mohammadi *et.al*¹⁰. The contradiction between our results and those of the

cited references are attributed to the difference in adsorption separation (for this study) and membrane separation (in References 10 and 11). The pervaporation separation using membrane filters is related to numerous factors such as gas permeation or adsorption to the solid, the gas diffusion in the solids owing to the difference in gas pressure or concentration, and the gas desorption from the solid. Conversely, the adsorption separation includes only one of these steps and the most important factor is the interaction between gas molecules and the solids. Furthermore, pervaporation is conducted based on high temperature (*i.e.* c.a.373 K in ref.10) to permeate gas molecules to solid more rapidly even when the adsorption separation of water isotopes was conducted at 298 K. Generally, the quantum effects of water isotopes are significant at ambient temperature and vanish with increasing temperature. At this point, D₂O selective filtration by pervaporation may occur primarily because of the kinetic mechanism underlying heavier molecules of D₂O and not because of quantum effects.”

Furthermore, to confirm the H₂O selective adsorption characteristics of GO, we carried out an additional experiment on the mixed gas adsorption using the mixed vapor of D₂O and H₂O with the aid of newly constructed laboratory-designed equipment with a mass spectrometer to show the H₂O selective adsorption behavior of GO (FigS2 (a)).

However, the complex comparison between single component adsorption and mixed vapor adsorption must be avoided owing to the predominantly formed HDO in the H₂O-D₂O mixed phase because the chemical exchange occurs based on the following equilibrium constant:

$$K = \frac{[\text{HDO}]^2}{[\text{H}_2\text{O}][\text{D}_2\text{O}]}$$

$$= 3.85 \text{ (at 298 K)}$$

Therefore, we determined the H/D ratio of water isotopic mixtures using mass intensity measurements and compared it with the H/D value obtained from single component adsorption measurements. The vapor of the bulk mixture of H₂O: D₂O with a mole ratio of 1.13 :1.00 (*i.e.*, H/D = 1.13) was adsorbed by the GO at 298 K and $P/P_0 = 0.87$ for 24 h. We determined the composition of the mixed adsorbed layer on the GO in the liquid state as follows: we desorbed the adsorbed mixed layer on GO at 333 K for 2 h, collected the desorbed water isotopic mixture using a cold trap at 77 K, then converted it to liquid at 298 K, for which the H/D ratio was determined by mass spectroscopy. Fig. 2S(b) shows the time courses of the $m/z = 18$ to $m/z = 20$ intensity-ratios (I^{18}/I^{20}) for the feed vapor (red) and the desorbed vapor after adsorption by GO. The I^{18}/I^{20} of the adsorbed mixture

clearly exceeds that of feed vapor, indicating the relative high H contents in the adsorbed layer on GO. We determined the H/D ratios from the averaged I^{18}/I^{20} values for the last 100 s of these curves. The H/D ratios are 1.13 for the initial mixed vapor and 1.70 for the adsorbed layer, respectively, indicating selective adsorption of H₂O on GO.

The corresponding adsorption amounts of H₂O, D₂O, and HDO at $P/P_0 = 0.87$ were 8.36, 2.91, and 9.68 mmol/g with $K = 3.85$ at 298 K, respectively. The adsorption amounts of H₂O and D₂O at $P/P_0 = 0.87$ for the adsorption branch of the single component adsorption isotherm are 11.5 and 9.48 mmol/g, respectively. This corresponds to H/D = 1.21 being a smaller isotopic difference than that observed for the mixed vapor adsorption (Table S1). This increase in the H/D ratio in mixed vapor adsorption indicates that the D₂O-phobisity of GO is emphasized by the presence of H₂O compared to that from single component adsorption because of the competitive adsorption on the unfavorable adsorption site of oxygen functional groups for D₂O.

Fig.S2 H₂O selective adsorption experiment of isotopic mixed water vapors on GO |
(a) Experimental setup for the isotopic mixture adsorption on GO using mass spectroscopy. **(b)** The time courses of the mass spectra for the feed vapor (red) and the desorbed vapor after adsorption on GO at $P/P_0 = 0.87$ for 1days (blue). The intensity ratios for $m/z = 18$ and 20 are shown.

Table S1 Adsorption amounts of water isotopes on GO in the single-component and mixed vapor adsorption

	Mixed vapor adsorption	Single component adsorption
H ₂ O (mmol/g)	8.36	11.5
D ₂ O (mmol/g)	2.91	9.48
HDO (mmol/g)	9.68	-
H/D	1.70	1.21

REVIEWER COMMENTS

Reviewer #1 (Remarks to the Author):

The revised paper provides more data and explanation about the selective adsorption of H₂O by staggered GO structure, the related discussion is supported by the data. There are following points to be improved:

1.The title uses the term of "water isotope", which is not very clear, it can be H₂O/D₂O, or hydrogen isotopic water, as H₂¹⁶O/H₂¹⁸O are not covered in this work.

2.The work shows that staggered GO structure, or "staggered stacking structure of GO" as used in the abstract, can demonstrate selective adsorption of H₂O/D₂O, but "simply stacked structure of GO sheets" as used in Figure 1, or single GO sheet can not. In the paper, the term of "GO" sometimes is directly used to refer to staggered GO structure, such as Page 3, line 3, Page 2, line 30-31, "the GO and rGO", please recheck the whole paper to ensure that correct terms about "GO" are used for different cases.

3.Please clearly describe how the data in Page 4, line 32-35 and Table S1 were collected, if they were from Figure 3 or Fig. S6, please describe clearly. These data are very important. Otherwise, Fig. S2b can be explained from another perspective based on the reference J. Phys. Chem. C. 124 (2020) 26864-26873 : D₂O is absorbed by GO stronger, and D₂O desorption under the conditions of this work is more difficult, so the ratio in Fig. S2b changed for feed vapour and desorbed vapour.

4.As rGo-related work is done, it is worth adding the work in the abstract and conclusion part.

5.There are minor language points, "D₂O-phobosity", "D₂O-phobocity" "isotope effect", " isotopic effect", please check and use the same term for the work to be clear.

Reviewer #3 (Remarks to the Author):

The authors carefully addressed all my comments and I think this work will be a valuable addition to the literature. That said, there are still some aspects of the work that needs future works, most notably, as mentioned in the paper, "The contradiction between our results and those of the cited references are attributed to the difference in adsorption separation (for this study) and membrane separation (in References 10 and 11)."

So, I recommend this paper for publication without need for any further modification.

We are deeply grateful to all reviewers and have carefully considered their comments and suggestions for revisions to our manuscript.

Please find below our responses (in blue) to each of the reviewer comments (in black). The revised parts are presented in red.

COMMENTS by Reviewer #1

The revised paper provides more data and explanation about the selective adsorption of H₂O by staggered GO structure, the related discussion is supported by the data. There are following points to be improved:

Answer

We appreciate the valuable comments from Reviewer #1.

Comment (1)

The title uses the term of “water isotope” which is not very clear, it can be H₂O/D₂O, or hydrogen isotopic water, as H²¹⁶O/H²¹⁸O are not covered in this work.

Answer to comment (1)

Thank you for the comment regarding our ambiguously worded title. According to the reviewer’s suggestion, we have changed the title to “Staggered structural dynamic-mediated selective adsorption of H₂O/D₂O on flexible graphene oxide nanosheets.”

Comment (2)

The work shows that staggered GO structure, or “staggered stacking structure of GO” as used in the abstract, can demonstrate selective adsorption of H₂O/D₂O, but “simply stacked structure of GO sheets” as used in Figure 1, or single GO sheet can not. In the paper, the term of “GO” sometimes is directly used to refer to staggered GO structure, such as Page 3, line 3, Page 2, line 30-31, “the GO and rGO”, please recheck the whole paper to ensure that correct terms about “GO” are used for different cases.

Answer to comment (2)

According to the reviewer’s suggestion, we now distinguish between the two types of GO, where in the revised manuscript, “simply stacked ultra-large GO” refers to the generally accepted GO model and “staggered stacking structure of GO” refers to our proposed model.

In addition, we have carefully rechecked the main text of the manuscript as well as figure legends and supplemental information to avoid any unclear terminology.

Comment (3)

Please clearly describe how the data in Page 4, line 32-35 and Table S1 were collected, if they were from Figure 3 or Fig. S6, please describe clearly. These data are very important. Otherwise, Fig. S2b can be explained from another perspective based on the reference J. Phys. Chem. C. 124 (2020) 26864-26873 : D₂O is adsorbed by GO stronger, and D₂O desorption under the conditions of this work is more difficult, so the ratio in Fig. S2b changed for feed vapour and desorbed vapour.

Answer to comment (3)

We apologize for the unclear description related to mixed-gas adsorption data. The data presented on Page 4 in the manuscript were derived from the H₂O/D₂O mixed-gas adsorption experiment to obtain direct evidence of the preferential adsorption of H₂O on GO. We used a mass filter to determine the compositional changes of the H₂O/D₂O mixed-gas adsorption on GO. We prepared an H₂O/D₂O liquid mixture with a composition of H₂O:D₂O = 1.13:1.00 in mole fraction (i.e., H/D = 1.13) in a liquid reservoir equipped with an adsorption line. Herein, we mainly describe the composition of the mixture under the H/D ratio, as HDO molecules were present in the H₂O/D₂O mixture due to H/D exchange. In addition, we describe the hydrogen isotopic water mixture as an “H₂O/D₂O mixture,” even though the mixture contains HDO whose composition varies with the mixture composition of H₂O and D₂O at a constant temperature.

The temperature of the H₂O/D₂O liquid mixture in the liquid reservoir was kept constant at 297 ± 0.2 K, and the H₂O/D₂O mixed vapor was adsorbed on GO at 298 K for 1 d to achieve adsorption equilibrium at a relative vapor pressure of 0.94. Here, the relative vapor pressure for pure H₂O liquid was used, as the derivation of the relative vapor pressure of the H₂O/D₂O mixture is difficult due to H/D exchange.

We adsorbed H₂O/D₂O mixed vapor on GO at a near saturation vapor pressure, and sufficient amounts of the adsorbed mixture (i.e., approximately 90 mg) were collected in a cold trap by desorption to determine a reliable vapor composition as follows. Following adsorption equilibration and using weight loss measurements of GO adsorption of H₂O and D₂O under the desorption conditions of 333 K in vacuum (< 0.1 Pa) for 2 h (Table S4), the adsorbed H₂O and D₂O molecules were confirmed to have been completely desorbed from the GO. The desorbed H₂O and D₂O vapors were then

collected in a cold trap at 77 K for 2 h. The adsorption selectivity of GO for H₂O and D₂O was determined by measuring I^{18} and I^{20} , which are the mass intensities at $m/z = 18$ and 20, respectively. Because the observed compositions of H₂O and D₂O can provide the HDO content when the equilibrium relation is used, we show only the results for H₂O and D₂O.

The H/D ratio of the H₂O/D₂O mixture and the mass intensities were calibrated in advance (Fig. S13). As shown in Fig. S2(b), we measured the time courses of the mass spectra of the feed vapor and the vapor of collected H₂O/D₂O mixture adsorbed on GO due to the faster evaporation rate of lighter isotopes of H₂O in the early stages of the measurements. Then, we obtained reliable I^{18}/I^{20} values of the equilibrated composition of the vapors by averaging the curves over 100 s from 200 to 300 s. The preferential adsorption of H₂O on GO over that of D₂O was confirmed by the higher mass intensity ratio of I^{18}/I^{20} for the desorbed vapor ($I^{18}/I^{20} = 2.40$) than that for the feed vapor ($I^{18}/I^{20} = 1.25$).

Finally, the H/D ratio of the collected condensate of the adsorbed mixture was determined from the mass spectral data using the calibration curve shown in Fig. S13. The corresponding H/D ratios of the feed vapor and desorbed vapor after adsorption on GO were 1.13 and 1.70, respectively. Table S1 shows the H/D ratios evaluated from the mixed H₂O/D₂O adsorption measurements and single-component H₂O and D₂O adsorption isotherms on GO at $P/P_0 = 0.94$. Here, we evaluated the adsorbed amounts of H₂O, D₂O, and HDO on GO in a mixed adsorption experiment using the weight of the desorbed condensate (399 mg g⁻¹) and the H/D exchange constant of bulk water ($K = 3.94$ at 298 K). The corresponding H/D ratio of the single-component adsorption was 1.29, and the higher H/D ratio in the mixed-vapor adsorption (i.e., H/D = 1.70). This indicates that the D₂O-phobicity of GO was promoted because of selective H₂O adsorption even on the unfavorable adsorption sites of the oxygen functional groups for D₂O as compared with that of single-component adsorption.

Saidi et al. reported that D₂O can be strongly adsorbed on GO and D₂O desorption from GO is more difficult with their theoretical calculation for water molecules on GO sheets, evidencing the H₂O selective permeance on GO membrane in the pervaporation separation (P. Saidi et al., *J. Phys. Chem. C* 124 (2020) 26864–26873). However, in the present study, we confirmed higher adsorption amounts of H₂O than D₂O on GO for the entire range of P/P_0 by single-component adsorption isotherm measurements (see Fig. 3(a)) and mixed-vapor adsorption measurements (Fig. S2(b)). Thermogravimetric (TG) measurements and differential thermal analysis (DTA) of GO adsorbing water (see Fig. S14 and Fig. S15) indicated no clear difference in the desorption temperatures of

physisorbed H₂O and D₂O. The number of strongly adsorbed D₂O molecules must be considerably less than that of the total water adsorbed in a realistic GO structure. This is because, unlike the adsorption sites for H₂O, the adsorption sites for strongly adsorbed D₂O must be limited to only a part of the GO surface.

Lighter H₂O can permeate faster than D₂O through the GO membrane for pervaporation separation measurements, resulting in the observed higher H₂O content in the permeates, as reported by Saidi et al. Permeation is associated with non-equilibrium-adsorbed states of H₂O and D₂O, which are different from the adsorption states of H₂O and D₂O at equilibrium in our experiment. This difference must be elucidated in future studies.

In the SI, we provide details on the experimental setup and explain how we determined the adsorption amounts of H₂O, D₂O, and HDO on GO from our mixed-vapor adsorption measurements. This was necessary to clarify the difference between the method of mixed H₂O/D₂O vapor adsorption on GO and the previously reported pervaporation separation method with a GO membrane. In the supplemental information, we explain as follows:

“Using mass spectroscopy, we conducted adsorption measurements of the mixed vapor of H₂O and D₂O on GO to confirm the hydrogen isotopic water selective adsorption characteristics of GO. Figure S2(a) shows a schematic of the lab-made vapor adsorption line equipped with a quadrupole mass spectrometer (M-101QA-TDF, Canon ANELVA Co.) and a turbomolecular vacuum pump (T-Station 75, Edwards Co.). A GO sample (0.221 g) was used to determine the composition of the desorbed water mixture. Water obtained by filtration of distilled water with a Milli-Q Reference water system (Millipore Japan Co.) was used as H₂O. Here, the content of deuterium in H₂O was negligible (143 ± 2 ppm, Y. Ono, R. Futamura, K. Kaneko et al., *J. Colloid. Interface Sci.* 508 (2017) 14–17). Deuterium oxide (99.8 atom% deuterated, Kanto Chemical Co., Inc.) was used as D₂O. GO was pretreated at 333 K under vacuum (< 0.1 Pa) for 2 h prior to the adsorption measurements. We prepared an H₂O/D₂O liquid mixture with a mole fraction of H₂O:D₂O = 1.13:1.00 (i.e., H/D = 1.13) in a liquid reservoir of the adsorption line. Herein, we describe the composition of the mixture using the H/D ratio, as the formation of HDO molecules inevitably occurs in the H₂O/D₂O mixture. We refer to the hydrogen isotopic water mixture as an “H₂O/D₂O mixture,” even though the mixture contains HDO whose composition varies with the mixture composition of H₂O and D₂O at a constant temperature.

The 15 g of liquid mixture that was used in this experiment for the mixed feed vapor was sufficiently large compared to the amount adsorbed (i.e., ~90 mg). Thus, the change in the liquid H₂O/D₂O composition due to adsorption was negligibly small. The liquid mixture in the liquid reservoir was frozen at 77 K with liquid N₂ prior to mixed-vapor adsorption and purified by removing the dissolved air through evacuation under melting conditions three times. The mixed-vapor adsorption of H₂O and D₂O was conducted near the saturation vapor pressure because the difference in the amount of adsorption between H₂O and D₂O in the single-component adsorption isotherms near the saturation vapor pressure (Fig. 3(a)) was sufficiently large for reliable measurement of the compositional change of H₂O and D₂O in the mixed-vapor adsorption. The temperature of the H₂O/D₂O liquid mixture in the liquid reservoir was kept constant at 297 ± 0.2 K, and the vapor was adsorbed on GO at 298 K for 1 d to achieve adsorption equilibrium at a relative vapor pressure of 0.94. Here, the relative vapor pressure for pure H₂O liquid was used because the derivation of the relative vapor pressure of the H₂O/D₂O mixture was difficult due to H/D exchange.

The adsorbed H₂O and D₂O molecules were completely desorbed from the GO at 333 K following adsorption equilibration, and the desorbed H₂O and D₂O vapors were collected in a cold trap at 77 K for 2 h. We confirmed the entire desorption of the two molecules from GO using weight loss measurements of GO adsorption of H₂O and D₂O under the desorption conditions of 333 K in vacuum (< 0.1 Pa) for 2 h (Table S4).

The adsorption selectivity of GO for H₂O and D₂O was determined by measuring I^{18} and I^{20} , which are the mass intensities at $m/z = 18$ and 20 , respectively. Because the observed compositions of H₂O and D₂O can provide the HDO content when the equilibrium relation is used, we only show the results for H₂O and D₂O.

The H/D ratio of the H₂O/D₂O mixture and the mass intensities were calibrated in advance (Fig. S13). As shown in Fig. S2(b), we measured the time courses of the mass spectra of the feed vapor and the vapor of collected H₂O/D₂O mixture adsorbed on GO due to the faster evaporation rate of lighter isotopes of H₂O in the early stages of the measurements. Then, we obtained reliable I^{18}/I^{20} values of the equilibrated composition of the vapors by averaging the curves over 100 s from 200 to 300 s. The preferential adsorption of H₂O on GO over that of D₂O was confirmed by the higher mass intensity ratio of I^{18}/I^{20} for the desorbed vapor ($I^{18}/I^{20} = 2.40$) than that for the feed vapor ($I^{18}/I^{20} = 1.25$). The H/D ratio of the H₂O/D₂O mixture was determined from a calibration curve of the mass spectral intensity ratio of I^{18}/I^{20} versus the H/D ratio of the H₂O/D₂O mixture (Fig. S13). The corresponding H/D ratios of the feed vapor and desorbed vapor after adsorption on GO were 1.13 and 1.70, respectively. This indicated a higher H

content in the adsorbed mixture on GO than in the feed vapor (i.e., the amount of H₂O is greater than that of D₂O in the adsorbed mixture).

Table S1 compares the amounts of H₂O and D₂O adsorbed on GO for single-component and mixed-vapor adsorption. Here, an unclear comparison between single-component and mixed-vapor adsorption should be avoided due to the predominantly formed HDO in the H₂O/D₂O mixture, as chemical exchange occurs based on the following equilibrium constant:

$$K = \frac{[\text{HDO}]^2}{[\text{H}_2\text{O}][\text{D}_2\text{O}]} \\ = 3.85 \text{ (at 298 K)}$$

Therefore, we determined the H/D ratio of hydrogen isotopic water mixtures using mass intensity measurements and compared it with the H/D value obtained from single-component adsorption measurements. The adsorption amount of the hydrogen isotopic water mixture on GO was 399 mg g⁻¹, as determined by the weight measurement of GO following mixed H₂O/D₂O vapor adsorption. The corresponding adsorption amounts of H₂O, D₂O, and HDO in the mixed-vapor adsorption were 8.36, 2.91, and 9.68 mmol/g, respectively. Here, H/D = 1.70 and the equilibrium constant ($K = 3.85$) for the isotopic exchange reaction were used to determine the adsorption amounts for each component. The adsorption amounts for single-component adsorption at $P/P_0 = 0.94$ were 14.1 and 10.9 mmol g⁻¹ for H₂O and D₂O, respectively, which were scaled by 0.5 to compare with those of the 1:1 mixed-vapor adsorption. The corresponding H/D ratio of single-component adsorption was 1.29, and the higher H/D ratio under mixed-vapor adsorption (i.e., H/D = 1.70). This indicates that the D₂O-phobicity of GO was promoted because of selective H₂O adsorption even on the unfavorable adsorption sites of the oxygen functional groups for D₂O as compared with that of single-component adsorption.

Saidi et al. reported that D₂O can be strongly adsorbed on GO, and D₂O desorption from GO is more difficult with their theoretical calculation for water molecules on GO sheets, evidencing the H₂O selective permeance on GO membrane during pervaporation separation (P. Saidi et al., *J. Phys. Chem. C* 124 (2020) 26864–26873). However, in this study, we confirmed higher adsorption amounts of H₂O over D₂O on GO for the entire range of P/P_0 by single-component adsorption isotherm measurements (see Fig. 3(a)) and mixed-vapor adsorption measurements (Fig. S2(b)). Thermogravimetric (TG) measurements and differential thermal analysis (DTA) of GO adsorbing water (see Fig. S14 and Fig. S15) indicated no clear difference in the desorption temperatures of physisorbed H₂O and D₂O. The number of strongly adsorbed D₂O molecules must be

considerably less than that of the total water adsorbed in a realistic GO structure. This is because the adsorption sites for strongly adsorbed D₂O must be limited to only a part of the GO surface as compared with the adsorption sites for H₂O.

Under pervaporation separation measurements, lighter H₂O can permeate faster than D₂O through the GO membrane, resulting in higher H₂O content in the permeates, as reported by Saidi et al. Permeation is associated with the non-equilibrium-adsorbed states of H₂O and D₂O, which, in our experiment, were different from the adsorption states of H₂O and D₂O under equilibrium. We cannot discuss the relationship between the permeation and adsorption of H₂O and D₂O onto GO using the present adsorption data. Thus, this difference must be elucidated in another study.”

In addition, we also clearly and concisely explain mixed-vapor adsorption in the revised manuscript as follows (Page 4, Lines 18–32):

“Moreover, the mixed-vapor adsorption of H₂O and D₂O was conducted using laboratory-designed equipment with a mass spectrometer (see Fig S2(a)) to confirm the H₂O selective adsorption of GO. Herein, we describe the composition of the mixture with the H/D ratio, as HDO molecules inevitably form in the H₂O/D₂O mixture. We refer to the hydrogen isotopic water mixture as an “H₂O/D₂O mixture,” even though the mixture contains HDO whose composition varies with the mixture composition of H₂O and D₂O at a constant temperature.

In this measurement, the vapor of the liquid mixture of H₂O:D₂O with a mole ratio of 1.13:1.00 (i.e., H/D = 1.13) was adsorbed on GO at 298 K and $P/P_0 = 0.94$ for 24 h. We determined the H/D ratio of the adsorbed H₂O/D₂O mixture on GO as follows. We fully desorbed the adsorbed H₂O/D₂O mixture on GO at 333 K for 2 h and collected the desorbed H₂O/D₂O mixture using a cold trap at 77 K. Mass spectral measurements of I^{18} and I^{20} , which are the mass intensities at $m/z = 18$ and 20, for the desorbed vapor were then conducted to determine the H/D ratio of the H₂O/D₂O mixture (see details of the experimental setups in the SI).”

Figure. S2| H₂O selective adsorption experiment of mixed H₂O/D₂O vapors on GO. (a) Experimental setup for H₂O/D₂O mixed-vapor adsorption on GO using mass spectroscopy. (b) Time courses of the mass spectra for the feed vapor (red) and desorbed vapor (blue) after mixed-vapor adsorption on GO at 298 K and $P/P_0 = 0.94$ for 1 d. Here, the relative vapor pressure is for pure H₂O. The mass spectral intensity ratios for $m/z = 18$ and 20 are shown.

Table S1. Amounts of water isotopes adsorbed on GO in single-component and H₂O/D₂O mixed-vapor adsorption

	Mixed vapor adsorption	Single component adsorption
H ₂ O (mmol/g)	8.36	14.1
D ₂ O (mmol/g)	2.91	10.9
HDO (mmol/g)	9.68	-
H/D	1.70	1.29

Table S4. Weights of GO samples before and after adsorption and desorption of H₂O and D₂O

	Before adsorption ¹⁾ (g)	After adsorption ²⁾ (g)	After desorption ³⁾ (g)	Ratio of weights after desorption versus before adsorption
H ₂ O	0.101	0.126	0.102	1.01
D ₂ O	0.100	0.130	0.100	1.00

1) GO was pretreated at 333 K for 2 h under a vacuum (< 0.1 Pa) before the weight measurement.

2) Water vapor was adsorbed on GO at 295 K and $P/P_0 = 0.9$ for 1 d.

3) Water adsorbed on GO was desorbed under at 333 K and < 0.1 Pa for 2 h.

The following measurements were conducted to confirm that the H₂O and D₂O that were adsorbed onto GO were sufficiently desorbed under the desorption conditions of heating at 333 K *in vacuo* for 2 h. The weight¹⁾ of the GO sample prior to water adsorption was determined by preheating it at 333 K *in vacuo* for 2 h. The pretreated GO sample was placed in a desiccator containing saturated KCl H₂O or D₂O solutions for 1 d at 295 K to reach adsorption equilibrium. Subsequently, the weight²⁾ of the GO sample with adsorbed H₂O or D₂O at the equilibrium adsorption amount was measured. Finally, the GO sample with adsorbed H₂O or D₂O was heated at 333 K *in vacuo* for 2 h to determine the weight³⁾ of the GO sample following desorption. The determined weights are listed in Table S4.

The results presented in Table S4 ensure that adsorbed H₂O and D₂O are both thoroughly desorbed under the desorption conditions of heating at 333 K *in vacuo* for 2 h. The desorption conditions were used under mass spectroscopy to determine the adsorbed amounts of H₂O and D₂O in the mixed-vapor adsorption experiment. However, the weight measurement of GO with adsorbed water in atmosphere showed a slight error because of the inevitable desorption of water, which was unlike that in the single-vapor adsorption isotherm measurements.

Figure. S13 Relationship between the H/D composition of the H₂O and D₂O vapor mixtures and the observed $m/z = 18$ and 20 intensity ratios (I^{18}/I^{20}) of the mixed vapors.

Figure. S14 | TG profiles for GO adsorbing H_2O (red) and D_2O (blue) and their derivatives (DTG) conducted under N_2 flow at 250 mL min^{-1} . The heating rate was 1 K min^{-1} .

Figure. S15 DTA profiles for GO adsorbing H_2O (red) and D_2O (blue) conducted under N_2 flow at 250 mL min^{-1} . The heating rate was 1 K min^{-1} . (Insets) Magnified figures of the DTA profiles around $330 \text{ K} < T < 350 \text{ K}$.

Comment (4)

As rGo-related work is done, it is worth adding the work in the abstract and conclusion part.

Answer for comment (4) from authors

As recommended by the reviewer, in the Abstract and Conclusion, we have added the following text related to rGO:

Abstract (Page 1, Lines 27–31)

“Although the comparison of MD simulation results for the generally accepted ultra-large GO sheet model and reduced GO model revealed that the flexible nature of the interlayer spacing could be attributed to the oxygen-rich sheet structure of GO, the simple stacking model of ultra-large GO sheets cannot well explain the observed swelling behavior of GO from XRD experiments.”

Conclusion (Page 8, Lines 7–13)

“In this work, we firstly showed a novel aspect of GO on water adsorption from MD simulations. Although the flexible nature of the interlayer spacing of GO could be attributed to its oxygen-rich sheet structure, which was not observed with rGO, the simple stacking model of ultra-large GO sheets, which is a generally accepted GO model, cannot adequately explain the swelling behavior of GO observed in the XRD experiments. Our MD simulations of the staggered stacking of flexible GO sheets can effectively explain the swelling nature of GO upon water adsorption.”

Comment (5)

There are minor language points, “D₂O-phobisity”, “D₂O-phobicity” “isotope effect”, “isotopic effect”, please check and use the same term for the work to be clear.

Answer for comment (4) from authors

We apologize for the errors in spelling. We have carefully rechecked the manuscript, including figure legends and all supplemental information, and have unified the explanation as “D₂O-phobicity” and “isotopic effect”

COMMENTS by Reviewer #3

The authors carefully addressed all my comments and I think this work will be a valuable addition to the literature. That said, there are still some aspects of the work that needs future works, most notably, as mentioned in the paper, "The contradiction between our results and those of the cited references are attributed to the difference in adsorption separation (for this study) and membrane separation (in References 10 and 11)."

So, I recommend this paper for publication without need for any further modification.

Answer for comment from authors

We appreciate the reviewer's recognition of the importance of our study. Further studies are indeed necessary to identify the reasons for the differences observed between adsorption and pervaporation separation.

REVIEWERS' COMMENTS

Reviewer #1 (Remarks to the Author):

The revised manuscript gives better presentation of the results. I recommend this paper for publication after the following modification:

1. Please recheck the terms in the whole paper, like Page 4, line 20 "H₂O selective adsorption of GO", does that mean "H₂O selective adsorption on GO"?
2. Please check the red texts of the revised manuscript, as they were copied from the response to the reviewers file, please modify so that the manuscript is more consistent, and recheck the language of the manuscript.
3. The TEM and Raman data show the lateral size of GO in the range of nm, with the presence of wrinkling GO layers to form a staggered structure. The revised manuscript uses the term of "ultra-large GO sheet", as the term of "ultra-large GO sheet" usually indicates the lateral size of GO sheet larger than 10 μm or even larger in the field. Please recheck the term in the paper, and tailor the modelling related work in the whole paper to be more precise.

We are deeply grateful to the reviewer. We have carefully considered his/her comments and suggestions for revision of our manuscript.

Please find below our responses (in blue) to each of the reviewer comments (in black). The revised parts are presented in red.

COMMENTS by Reviewer #1

The revised manuscript gives better presentation of the results. I recommend this paper for publication after the following modification:

Comment (1)

Please recheck the terms in the whole paper, like Page 4, line 20 “H₂O selective adsorption of GO”, does that mean “H₂O selective adsorption on GO”?

Answer to comment (1)

As followed by the suggestion from the reviewer, we rechecked the manuscript carefully and revised properly in that point.

Comment (2)

Please check the red texts of the revised manuscript, as they were copied from the response to the reviewers file, please modify so that the manuscript is more consistent, and recheck the language of the manuscript.

Answer to comment (2)

We carefully rechecked the manuscript and revised the explanation more consistently and correctly in the languages, including the parts which we previously revised. We showed the revised parts in red on the manuscript. For example, we add the following sentences in the manuscript to clarify the experimental method of the mixed vapor adsorption of H₂O/D₂O on GO (Page 4-5, Line 35-5):

“We confirmed the entire desorption of H₂O and D₂O molecules from GO using weight loss measurements of GO adsorbing water molecules under the desorption conditions of 333 K in vacuum (< 0.1 Pa) for 2 h (Table S4). Furthermore, we conducted thermogravimetric (TG) measurements and differential thermal analysis (DTA) of GO adsorbing water (see Fig. S14 and Fig. S15), and we confirmed there is no clear difference

in the desorption temperatures of physisorbed H₂O and D₂O.”

(Page 5, Line 23-33):

“Here, an unclear comparison between single component adsorption and mixed vapor adsorption should be avoided due to the predominantly formed HDO in the H₂O/D₂O mixture, as chemical exchange occurs based on the following equilibrium constant:

$$K = \frac{[\text{HDO}]^2}{[\text{H}_2\text{O}][\text{D}_2\text{O}]}$$
$$= 3.85 \text{ (at 298 K)}$$

Therefore, the determined H/D ratio of hydrogen isotopic water mixtures adsorbed on GO was compared with the H/D value obtained from single component adsorption measurements as follows.

The adsorption amount of the hydrogen isotopic water mixture on GO was 399 mg g⁻¹, as determined by the weight measurement following the mixed H₂O/D₂O vapor adsorption.”

Comment (3)

The TEM and Raman data show the lateral size of GO in the range of nms, with the presence of wrinkling GO layers to form staggered structure. The revised manuscript uses the term of “ultra-large GO sheet”, as the term of “ultra-large GO sheet” usually indicates the lateral size of GO sheet larger than 10 μm or even larger in the field. Please recheck the term in the paper, and tailor the modelling related work in the whole paper to be more precise.

Answer to comment (3)

Thank you for the comment. We believe the lateral size of GO sheet in the range of nms plays crucial roles in the realistic GO including the H₂O selective adsorption in the interlayer structure of GO as shown with our MD simulation. In molecular simulations, GO structures are generally modeled with an infinite length of oxidized graphene structures using periodic boundary conditions in the *x*- and *y*- directions (i.e. in-plane directions), which is same as our ultra-large GO simulations. This model is directly related to the GO structure whose lateral size is over 10 μm because of the fewer edges and the absence of the inter sheets junction structure, called “ultra-large GO” (Dong, L. Chen, Z. Lin, S. Wang, K. Ma, C. & Lu, H. Reactivity-Controlled Preparation of Ultralarge Graphene Oxide by Chemical Expansion of Graphite. *Chem. Matter.* **29**, 565-572 (2017); Lin, X. Shen, X. Zheng, Q. Yousefi, N. Ye, L. Mai, Y.-W. & Kim, J.-K. Fabrication of Highly-Aligned, Conductive, and Strong Graphene Papers Using

Ultralarge Graphene Oxide Sheets. *ACS Nano* **6**, 10708-10719 (2012)). Therefore, we called this model “ultra-large GO” in this study. To clarify, we rechecked whole paper for more precise explanation about this point including Method section and we added the following sentences in Page 16 of Lines 21-25:

“For the simply stacked GO model, four GO sheets were put in a simulation cell ($L_x = 4.263$ nm, $L_y = 3.938$ nm, and $L_z = 2.8$ nm) with an interlayer distance of 0.7 nm in the z -axis direction. Each GO sheet was alternately shifted by half the length of the simulation cell in the x - and y -axis directions and periodic boundary conditions were applied in all three directions. Since the lateral size of this GO model can be regarded as an infinite length due to the periodic boundary conditions, this calculation with the model is directly related to the important application of ultra-large GO sheets whose lateral size is over 10 μm with fewer edges and lower inter sheets junction structure. Therefore, we call this GO model “ultra-large GO” in this study.”